

# Distribution and drivers of symbiotic and free-living

# diazotrophic cyanobacteria in the Western Tropical South

# Pacific

Marcus Stenegren[1], Andrea Caputo[1], Carlo Berg[2], Sophie Bonnet[3,4], Rachel A Foster[1]

[1] Stockholm University, Department of Ecology, Environment and Plant Sciences. Stockholm, Sweden

[2] Science for Life Laboratory/Department of Biology and Environmental Science, Linnaeus University, Kalmar, Sweden

[3] Aix Marseille Université, Centre National de la Recherche Scientifique. Marseille/Noumea, New Caledonia, France

[4] Mediterranean Institute of Oceanography, Institut de Recherche pour le Dévelopement. Marseille, France

*Correspondence to*: Marcus Stenegren (marcus.stenegren@su.se)



**Abstract**
The abundance and distribution of cyanobacterial diazotrophs were quantified in two regions
(Melanesian archipelago, MA and subtropical gyre, SG) of the Western Tropical South
Pacific using *nifH* qPCR assays. UCYN-A1 and A2 host populations were quantified using
18S rRNA qPCR assays including one newly developed assay. All phylotypes were detected
in the upper photic zone (0-50 m), with higher abundances in the MA region. *Trichodesmium*
and UCYN-B dominated, composing 81-100% of *nifH* copies detected. Het-1 was the next
most abundant, and co-occurred with het-2 and het-3. The two UCYN-A lineages were least
abundant (<1.0-1.5 % of total *nifH* copies) and poorly detected (>47%). Abundance of the
UCYN-A hosts mirrored their respective symbionts; UCYN-A1 and A2 however were
detected while their respective hosts were below detection, suggesting a lower partner fidelity
or free-living life history. Pairwise comparisons of abundance and environmental parameters
supported two groups: deep (45 m) comprised of UCYN-A and surface (0-15m) comprised of
*Trichodesmium*, het-1 and het-2, while UCYN-B overlapped. Temperature, salinity, and PAR,
were positively correlated with the latter abundances except UCYN-A. Similar results were
identified in a meta-analysis of 11 external datasets. Combined, our results indicate that
conditions favoring the UCYN-A symbiosis differ from those of diatom diazotroph
associations and free-living diazotrophs.



## 1 Introduction

Biological di-nitrogen ($N_2$) fixation is considered a major source of new nitrogen (N)
to oceanic ecosystems (Karl et al., 1997). $N_2$ fixation is an energetically expensive process,
where $N_2$ gas is reduced to bioavailable ammonia (Howard and Rees, 1996) and is performed
by a small but diverse group of bacteria and archaea. The nitrogenase enzyme, which is
encoded by a suite of *nif*-genes, mediates $N_2$ fixation (Jacobson et al., 1989; Young, 2005).
Nitrogenase has a high iron (Fe) requirement (Howard and Rees, 1996), and often $N_2$ fixers,
or diazotrophs, are Fe- limited (Kustka et al., 2003; Raven, 1988). Nitrogenase is also
sensitive to oxygen ($O_2$), which has been shown to negatively influence $N_2$ fixation efficiency
(Meyerhof and Burk, 1928; Stewart, 1969). Thus, autotrophic diazotrophs (e.g. cyanobacteria)
have evolved strategies, such as temporal and spatial separation of the fixation process, to
protect their nitrogenase from $O_2$ evolution during photosynthesis (Berman-Frank et al., 2001;
Haselkorn, 1978; Mitsui et al., 1986). $N_2$ fixation is widespread and occurs in marine, limnic
and terrestrial habitats. In marine ecosystems it mainly occurs in the photic zone, closest to
the surface, however, more recently, evidence has shown activity in deeper depths below the
photic zone, including oxygen minimum zones (Benavides et al., 2016; Bonnet et al., 2013;
Fernandez et al., 2011; Halm et al., 2009; Löscher et al., 2015).
$N_2$ fixation in the photic zone is often attributed to a diverse group of cyanobacteria.
Traditionally, marine, photic dwelling diazotrophs are divided into two groups based on cell
diameter, e.g. > 10 μm and < 10 μm size fractions. Diatom diazotroph associations (DDAs),
symbioses between heterocystous cyanobacteria and a variety of diatom genera and large
filamentous non-heterocystous *Trichodesmium* spp., compose the larger size fraction (>10
μm). *Trichodesmium* spp. occurs as free filaments or often in two morphologies of colonies:
tufts/rafts and puffs. There are three defined lineages of the symbionts of DDAs based on their
*nifH* phylogeny: het-1 and het-2 refers to the two the *Richelia intracellularis* lineages which



associate with diatom genera, *Rhizosolenia* and *Hemiaulus,* respectively, while the third
lineage, het-3, is a symbiosis between the heterocystous *Calothrix rhizosoleniae* and
*Chaetoceros compressus* diatoms (Foster et al., 2010; Foster and Zehr, 2006).

4       The unicellular diazotrophic cyanobacterial groups are divided into: UCYN-A,

UCYN-B, and UCYN-C groups and are representatives of the <10 μm size fraction. The
UCYN-A (*Candidatus Atelocyanobacterium thalassa*) group can be further delineated into 4
sub-clades (lineages), two (UCYN-A1, UCYN-A2) are identified as symbiotic with small
prymnesiophyte microalgae (reviewed by Farnelid et al., 2016, see references within). The
UCYN-B group has its closest cultured relative as *Crocosphaera watsonii* and lives freely,
colonially, and also in symbiosis with the diatom *Climacodium frauenfeldianum* (Bench et al.,
2013; Carpenter and Janson, 2000; Webb et al., 2009; Zehr et al., 2001). Often overlooked, is
the observation that UCYN-B, when colonial or symbiotic could also be associated with the >
10μm size fraction. Less is known about the UCYN-C, and given that its *nifH* nucleotide
sequence is 90% similar (Foster et al., 2007) to *Cyanothece* spp. ATCC51142, it is assumed to
be analogous, and thus co-occur with the other < 10 μm size fraction. A diverse group of free-
living heterotrophic bacteria (e.g. gamma proteobacteria) (Berthelot et al., 2015; Bombar et
al., 2016; Halm et al., 2012; Langlois et al., 2005) and archaea (Zehr et al., 2005) are also
within the < 10 μm size fraction.
The Tropical South Pacific Ocean (TSP) is considered one of the most oligotrophic
regions in the World's oceans (Claustre and Maritorena, 2003) with a widespread N
deficiency (Deutsch et al., 2007; Raimbault et al., 2007) and in the central SP gyre, some of
the lowest concentrations of dissolved Fe in the world have been reported (Blain et al., 2008).
One exception is the Western Tropical South Pacific (WTSP), harboring many islands with Fe
rich sediments adding to an island mass effect (Shiozaki et al., 2014) and being influenced by
multiple ocean currents, both surface and subsurface, that drive the distribution of dissolved



nutrients, micronutrients, and the biota (Fitzsimmons et al., 2014; Gourdeau et al., 2008;
Marchesiello and Estrade, 2010; Wells et al., 1999). The structure of these currents also
promotes shearing instabilities and strong eddies (Qiu et al., 2009). Moreover, Van Den
Broeck et al. (2004) suggested that the WTSP is phosphate limited, while Law et al. (2011)
hypothesized that primary production and $N_2$ fixation in the WTSP follows the seasonality of
cyclones, which in their wake, enrich surface waters with phosphate, and fuel primary and
new production. An earlier investigation along a transect in the western equatorial Pacific
estimated that 74% of the total $N_2$ fixation could be attributed to the <10 µm size fraction as
abundances of unicellular cyanobacteria were high (17 cells $mL^{-1}$) (Bonnet et al., 2009).
However, diazotroph quantification is lacking further South in tropical waters, despite being
recently recognized as a hot spot of $N_2$ fixation, with average rates of ~570 µmol N $m^{-2}$ $d^{-1}$
(Bonnet et al., this issue), i.e. in the upper range (100-1000 µmol N $m^{-2}$ $d^{-1}$) of rates gathered
in the global $N_2$ fixation MAREDAT database (Luo et al., 2012).

14       The distribution and activity of diazotrophs in open ocean ecosystems are governed by

different ambient environmental factors, including macronutrient availability (Moutin et al.,
2008; Sañudo-Wilhelmy et al., 2001) and temperature (Messer et al., 2016; Moisander et al.,
2010). There are also simultanous influences by several factors (i.e. co-limitation of nutrients,
Mills et al., 2004). Moreover, most oceanic models of $N_2$ fixation assume that all diazotrophs
are equally controlled by the same environmental parameters (Deutsch et al., 2007; Hood et
al., 2004; Landolfi et al., 2015), despite well recognidzed differences in genetic repertoires for
assimilating dissolved nutrient pools (e.g. dissolved organic phosphate, Dyhrman et al., 2006;
Dyhrman and Ruttenberg, 2006), life histories (free, symbiotic, colonial), and cell sizes (µm
to mm).

24       The primary aim of this study was to quantify diazotroph abundance and distribution in

the WTSP with an emphasis on symbiotic $N_2$-fixing populations; both by ´at sea´ and lab



based quantitative approaches. For a more comprehensive investigation of the symbiotic
diazotrophs we developed a new primer and probe set for quantification of the UCYN-A1
host. We also identified key environmental parameters, both biotic and abiotic, which
influenced the distribution of diazotrophs in the WTSP and tested the congruency of these
parameters in an additional 11 publicly available datasets. We hypothesized that the
distribution and the underlying factors of the diazotrophic symbioses should differ due to the
major differences in host taxonomy (e.g. diatom vs. prymnesiophyte), size (1-2 μm to 100's
μm), and life history (free vs. symbiotic; chain forming). For comparison and for similarly
divergent characteristics (symbiotic vs. free; colonial vs. single), several free-living (UCYN-
B, *Trichodesmium* spp.) cyanobacterial diazotrophs were also included.

## 2 Materials and Methods

### 2.1 Sampling

Sampling was conducted on a transect in the WTSP during austral summer (19 Feb-5 Apr,
2015), on board the R/V *L'Atalante* (Fig. 1a). Nucleic acid samples were taken from 18
stations: three long duration (LD A, B and C) stations (approximately eight days duration) and
15 short duration (SD 1-15) stations (approximately eight hours duration). The cruise transect
was divided into two geographic regions (Fig. 1a). The first region (Melanesian archipelago,
MA) included SD 1-12, LD A and LD B stations (160º E-178º E and 170º-175º W). The
second region (subtropical gyre, SG) included SD 13-15 and LD C stations (160º W-169º W).
LD stations were chosen based on hydrographic conditions, satellite imagery, microscopic
analyses of >10 μm cyanobacterial diazotrophs and the results of 'at sea' qPCR analyses of
four unicellular diazotrophic targets (UCYN-A1, UCYN-A2, UCYN-B and UCYN-C) (see
below and Moutin et al., this issue). Seawater (2.5 L) was collected into clean (10% bleach

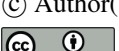



rinsed) 2.75 L polycarbonate bottles from 6-7 discrete depths based on surface incident light
intensity (100, 75, 54, 36, 10, 1, and 0.1%) once per station at both SD and LD stations using
Niskin bottles (12 L) arranged on a Conductivity Temperature Depth (CTD) rosette.

4         After collection from the CTD rosette, seawater was immediately filtered onto a 0.2 µm

pore size Supor filter (Pall Corporation, Pall Norden AB, Lund, Sweden) held within a 25 mm
diameter swinnex filter holder (Merck Millipore, Solna, Sweden) using a peristaltic pump
(Cole-Parmer, Masterflex, Easy-load II, USA). The filters were placed in pre-sterilized bead
beater tubes (Biospec Bartlesville, OK, USA) containing 30 µL of 0.1 mm and 0.5 mm glass
bead mixture, flash frozen in liquid nitrogen and archived at -80 ºC. Four additional DNA
samples were collected from 4 discrete depths, (75, 50, 36, 10 % light), at 11 of the 18
stations, for the 'at sea' qPCR (see below) and filtered as described above.
**2.2 Nutrient analyses**
Seawater for nutrient analyses was collected from each station using the CTD rosette at the
same depths as those collected for the nucleic acids. Seawater for inorganic nutrient analysis
were collected in 20 mL high-density polyethylene HCL-rinsed bottles and poisoned with
$HgCl_2$ to a final concentration of 20 µg $L^{-1}$ and stored at 4°C until analysis. Dissolved nitrate
and nitrite ($NO_3^-$+$NO_2^-$, DIN), phosphate ($PO_4^{3-}$, DIP) and silicate (Si $(OH)_4$, DiSi)
concentrations were determined by standard colorimetric techniques using a segmented flow
analyzer according to Aminot and Kérouel (2007) on a SEAL Analytical AA3 HR system
(SEAL Analytica, Serblabo Technologies, Entraigues Sur La Sorgue, France). Quantification
limits for nitrate, phosphate and silicate were all 0.05 µmol $L^{-1}$.
*Cell abundances and microscopy observations.* At the LD stations, 5 L of seawater was
collected at the same depths and parallel with the nucleic acid samples from the CTD-rosette.
Two sets of samples, one set each day, were taken on two different days (day 1 and 3 at each





LD station) and immediately filtered onto a 47 mm diameter Poretics (millipore) membrane
filter with a pore size of 5 μm using a peristaltic pump.
At the SD stations, the same collection was implemented, however a 25 mm diameter Poretics
membrane filter was used. The high densities of cells on the latter made it impossible to
properly enumerate the various cyanobacterial diazotrophs and as such these samples were
used only for qualtitative observations (see below). Immediately after filtration, samples were
fixed in 1 % paraformaldehyde (v/v) for 30 min prior to storing at -20 ºC. For enumeration,
the filter was mounted on a glass slide and examined at under an Olympus BX60 microscope
equipped with a filter for blue (460-490 nm) and green (545-580 nm) excitation wavelengths.
Three areas (area = 0.94 mm$^2$) per filter were counted separately and values were averaged.
When abundances were low, the entire filter (area = 1734 mm$^2$) was observed and cells
enumerated. Due to weak fluorescence, only *Trichodesmium* colonies and free-filaments
could be accurately estimated by microscopy and in addition, the larger cell diameter
*Trichodesmium*, herafter referred to as *Katagynemene pelagicum*, was enumerated separately
as these were often present albeit at lower cell densities. Other cyanobacterial diazotrophs, e.g
*C. watsonii*-like, *C. rhizosoleniae* (het-3) and *R. intracellularis* (het-1, het-2) were also
present on the larger 47 mm diameter samples, however fluorescence was weak and therefore
difficult to enumerate. Pico-eukaryote populations, identified as round 1-3 μm diameter cells,
with red excitation under the blue filter set, were also observed. For the latter populations,
qualitative observations of presence and some details on cell integrity (e.g. fluorescence,
frustule, free-living or symbiotic form) are included.
**2.3 DNA extraction**
The DNA from the 120 archived samples was extracted as described in Moisander et al.
(2008), with a 30 second reduction in the agitation step in a Fast Prep cell disrupter (Thermo,
Model FP120; Qbiogene, Inc. Cedex, France) and an elution volume of 70 μL. The nucleic



acid samples collected for the 'at sea' qPCR were extracted immediately after filtration using
a modified version of the DNAeasy plant kit (Qiagen) total DNA extraction protocol. The
modifications were an initial 2-minute agitation step using a bead beater (Biospec
MiniBeadBeater-16, Model 607EUR; Biospec) and final elution volume was 25 μl.
**2.4 Oligonucleotide design**
A new primer and probe set was designed to amplify the UCYN-A1 host and was based on
published 18S rRNA sequence (accession number JX291893) reported from N. Pacific gyre
(station ALOHA) (Thompson et al., 2012). The design utilized the same 96 bp target region of
the 18S rRNA used to amplify UCYN-A2 hosts described in Thompson et al. 2014 (Suppl.
Table 1). The primers and probe for the UCYN-A1 host 18S rRNA gene assay are as follows:
Forward, 5' AGGTTTGCCGGTCTGCCGAT-3'; Reverse, 5'
GAGCGGGTGTCGGAGACGGAT-3'; Probe, 5'-FAM-CTGGTAGAACTGTCCT-
TAMRA-3'. The forward, reverse and probe contain 2-4, 1, and 5 mismatches, respectively,
to UCYN-A2 host sequences (accession number KF771248-KF771254) and the following
closely related sequences (98-100%): uncultured eukaryote clones (station ALHOA:
EU50069; Cariaco Basin: GU824119) *Chrysochromulina parkeae*: AM490994),
*Braarudospaera bigelowii* TP056a: AB250784 *B. bigelowii* Furue-15: AB478413; *B.*
*bigelowii* Funahama T3: AB478413; *B. bigelowii* Yastushiro-1 AB478414. The UCYN-A1
oligonucleotides specificity was tested *de nova* against the following closely related sequences
derived from uncultured eukaryotic clonal sequences (accession numbers: EU500067-68;
FJ537341; EU500138-39; EF695227; EU500141; EU499958; EF695229; EF695220). Only
one mismatch was found in the forward probe for one sequence (EU500138). Finally, a cross
reactivity test between the newly designed UCYN-A1 host oligonculeotides and a dilution
series of the UCYN-A2 host template was run (see below).



**2.5 Quantitative PCR**
Abundances of selected diazotrophs *nifH* gene copies (UCYN-A1, UCYN-A2, UCYN-B,
UCYN-C, het-1, het-2, het-3 and *Trichodesmium* spp.) and the 18S rRNA of UCYN-A1 and
A2 hosts were performed using previously published oligonucleotides and TaqMAN assays
(Church et al., 2005; Foster et al., 2007; Moisander et al., 2010; Thompson et al., 2014) and
the newly designed UCYN-A1 host oligonculeotides (Suppl. Table 1). The qPCRs were
conducted in a StepOnePlus system (Applied Biosystems, Life Technologies, Stockholm
Sweden) in fast (>40 min) mode with the following parameters: 95 ºC for 20 s, followed by
45 cycles of 95 ºC for 1 s and 60 ºC for 20 s.
Cross reactivity tests were run on two of the heterocystous symbiont (het-1 and het-2)
oligonucleotides, the UCYN-A1 and UCYNA-2 oligonucleotides, and the newly designed
UCYN-A1 host oligonucleotides and UCYN-A2 host primer and probe set. The standard
curve for a particular target was run in reactions with the other primers and probe sets. For
example, the UCYN-A1 TaqMAN host primers and probes were run in reactions with UCYN-
A2 template DNA. The cross reactivity for the het-1 and het-2 primer and probe sets has been
previously reported (Foster et al. 2007), however only when the assay is run in standard mode.
Standard mode runs the holding, denaturation and annealing stages at the following longer
intervals than in Fast mode: 11 min and 40 s, 14 s, and 40 s, respectively. Hence, we tested the
cross-reactivity for the het primers and probes when run in fast mode, as the fast mode was
used in our study. Similarly, the cross-reactivity between UCYN-A1 and UCYN-A2 were
tested in fast mode at two annealing temperatures 60 ºC and 64 ºC; 64 ºC is the recommended
annealing temperature for the UCYN-A2 assay (Thompson et al. 2014).
Reaction volume was 20 µL in all qPCRs and consisted of 10 µL of 2X TaqMan fast buffer
(Applied Biosystems, 5.5 µL of nuclease free water (Sigma Aldrich Sweden AB, Stockholm
Sweden), 1 µL each of the forward and reverse primers (10 µM), 0.5 µL of fluorogenic probe



(10 µM) and 2 µL of DNA extract. For standard mode runs, the latter master mix was
identical with the exception of replacing the fast 2X buffer with the standard 2X buffer. For
reactions quantifying *Trichodesmium* spp. *nifH* copies, SD 9 was excluded and 1 µL of DNA
template was used for the remaining stations due to low template volume, and total reaction
volume was adjusted by addition of 1 µL of nuclease free water. Reactions were performed in
duplicates for the 'at sea' qPCR and in triplicates for the archived samples and lab based
qPCR. For the 'at sea' qPCR, only four targets (UCYN-A1, UCYN-A2, UCYN-B, and
UCYN-C) were quantified and only at the SD stations. No assays were processed at SD 5-6,
10-12, and 14 for the 'at sea' qPCR. Two µL of nuclease free water was used as template in
no template controls (NTCs); no *nifH* copies were detected in the NTCs.
Gene copy abundance was calculated from the mean Ct value of the 3 replicates and the
standard curve for the appropriate oligonucleotides in the lab based qPCRs. For the 'at sea'
qPCR, a mean Ct value of 2 replicates was used to maximize the number of samples run on
one amplification plate (96 well). In samples where 1 or 2 out of 3 replicates produced an
amplification, signals were noted as detectable, but not quantifiable (dnq) and no
amplification was noted as below detection (bd).

**2.6 Standard curves and PCR efficiency**

Standard curves were plotted and analyzed in Excel for each target based on the qPCR cycle
threshold (Ct) values from known dilutions of synthesized target gene fragments (gBlocks®;
Integrated DNA Technologies, Leuven Belgium) (359 bp *nifH* and 733 bp 18S rRNA for
UCYN-A hosts). Tenfold dilutions were made starting with $10^8$ to $10^1$ gene copies $L^{-1}$. The
PCR efficiency was determined as previously described (Short et al., 2004) for 12 samples run
on the het-1, het-2, and het-3 primers and probe tests. The qPCR efficiency ranged from 90-
99 % with an average of 94 % efficiency for the diazotroph targets het-1, het-2 and het-3.



**2.7 Statistics and data analysis**
Skewness and normal distribution tests by descriptive statistics was performed in IBM SPSS
(ver. 23) on the following parameters recorded during sample collection in the WTSP from
the CTD package: depth (m), oxygen (ml $L^{-1}$), temperature (ºC), chlorophyll fluorescence (μg
$L^{-1}$), photosynthetically active radiation (PAR; μmol photons $m^{-2}$ $s^{-1}$), salinity (PSU), and gene
copy abundances determined by qPCR. Significant skew was noted when skewness, divided
by its standard deviation, exceeded 1.95. All but three targets (het-1, UCYN-B and
*Trichodesmium* spp.) and three environmental parameters (temperature, salinity and oxygen)
were significantly skewed (not normally distributed) even after LOG10 transformation.
Therefore a non-parametric Spearman's rank correlation was conducted to test possible
correlations between the targets and environmental parameters, where we assume that the het
groups and UCYN-A clade is symbiotic, while UCYN-B is free living. The resulting
correlation matrices were visualized in the form of a heat map of hierarchical clustering in R
(ver. 3.2.2) using packages 'hmisc' and 'gplots'. Multivariate statistics by redundancy
analysis (RDA) was conducted using the R package 'vegan'. T-tests, in IBM SPSS (ver. 23)
were performed to characterize the different regions along the cruise transect based on
environmental parameters, including nutrients, measured between stations and was reported as
mean concentrations. For meta-analysis on the external dataset from 11 publically available
datasets, sampled in the Atlantic, Pacific and South China Sea, data was acquired from the
PANGAEA database and previous publications (Benavides et al., 2016; Bombar et al., 2011;
Church et al., 2005, 2008, Foster et al., 2007, 2009; Goebel et al., 2010; Kong et al., 2011;
Langlois et al., 2008; Moisander et al., 2008, 2010). We included only datasets with a
minimum of 10 datapoints on the previously mentioned diazotrophic targets. Note that in all
datasets the two UCYN-A phylotypes (A1 and A2) were not distinguished, and het-3 was
excluded since it was rarely quantified. The meta-analysis was conducted using the software





OpenMEE (based on R package 'metafor'), where correlation coefficients from Spearman's
rank were z-transformed (Fisher's) and tested using weighted random effect models.
Graphical visualization of the mean abundances of the most numerous diazotrophs across the
cruise transect was also performed in IBM SPSS (ver. 23).
**3 Results**
**3.1 Hydrographic conditions**
Near surface (0-5m) DIN concentrations were below the quantification limit (bq) in both the
MA and SG regions, while the mean surface DIP and DiSi concentrations were below the
quantification limit or low across all stations in the MA (bq-0.08 μM and 0.54-0.56 μM,
respectively) and significantly ($p<0.001$; t-test) higher ($0.18 \pm 0.07$ μM and $0.79 \pm 0.04$ μM,
respectively) at the stations in the SG (Table 1). The upper 25-30 m of depth had stable
temperatures of 29-30 ºC. The depth of the deep chlorophyll maximum (DCM) was between
70-165 m, except for LD B (DCM at 35 m), which was sampled during a degrading surface
phytoplankton bloom, and a 30-day composite of the surface chlorophyll *a* (Chl *a*) confirmed
the decreasing level of surface fluorescence measured by the CTD package at LD B (data not
shown).
**3.2 Comparison of 'at sea' and lab-based qPCR**
In order expedite the sample processing for the 'at sea' qPCR, a shortened and modified DNA
extraction protocol was performed, 4 depths were sampled, and 4 targets run (UCYN groups).
In total, 44 samples can be compared with results from the parallel archive samples and we
considered only when there was at least one order of magnitude difference in detection. A
summary of the comparison, including the difference in *nifH* copy abundance is provided in
Suppl. Table 2.



In general, the 'at sea' and lab based qPCR were similar in quantifying the targets.
Discrepancies were noted in 7, 8 and 11 samples, which had higher detection in the 'at sea'
analyses for UCYN-A1, UCYN-A2 and UCYN-B, respectively. There were fewer instances
(3, 4, and 5, respectively) of samples processed in the lab with the full extraction that had
higher abundances for the UCYN-A1, UCYN-A2 and UCYN-B, respectively.
*Horizontal and vertical distributions*. *Trichodesmium* and UCYN-B were the most abundant
diazotrophs and abundances ranged $10^4$-$10^6$ *nifH* copies L$^{-1}$ at multiple depths (4-6 depths) in
the upper water column (0-35 m) (Fig. 1-2; Suppl. Tables 3). *Trichodesmium* represented 80-
99% of total *nifH* genes detected at 9 out of 17 stations with highest detection in the MA and
low to undetected in the SG. Microscopy observations and abundances of *Trichodesmium* spp.
confirmed a high abundance of free filaments of *Trichodesmium* and *C. watsonii*-like cells at
LD B, while colonies were in general rarely observed (Suppl. Table 5).
At stations where *Trichodesmium* was not the most abundant diazotroph (e.g. SD 2, 6,
7, 14, 15, and LD C), UCYN-B had the highest depth integrated *nifH* copy abundance.
UCYN-B was also the most consistently detected diazotroph, and was quantifiable from all
stations sampled accounting for for 81-100% of the total detected *nifH* gene copies in the SG.
There was also a depth dependency for maximum abundance such that the average depth
maximas of *Trichodesmium* and UCYN-B at the stations in the MA were 10 and 25 m,
respectively. In the SG, the average depth maximum was the same for UCYN-B (25 m), while
the average depth of the *Trichodesmium* maximum deepened to 31m.
Of the three heterocystous cyanobacterial symbiont lineages (het-1, het-2, het-3), het-1
was the most dominant (60% detection in total samples, 72 of 120 samples), and similar to
*Trichodesmium*, had higher detection in the stations of the MA region. For example, at
stations SD 2, 4 and 9, het-1 represented 10-15% of the total *nifH* genes quantified in the
depth profiles, but in the total *nifH* genes quantified across the entire transect, het-1 only





represented 1.5 %. Abundances for het-1 ranged between $10^3$-$10^5$ *nifH* copies L$^{-1}$ (15 of the 18
stations) at multiple depths (0-90 m) and the average depth maximum at MA stations was
closer to the surface (15 m) compared to the SG stations (60 m) (Fig. 1; Suppl. Table 3). Het-
2 and het-3 co-occurred with het-1, however at lower abundances ($10^2$-$10^4$ *nifH* copies L$^{-1}$)
and unlike het-1, were bd at all depths sampled in 1 and 3 stations, respectively, located in the
SG. The averge depth of maximum abundance (17 m) for het-2 was similar to het-1 (15 m),
while het-3 was deeper at 33 m (considering only the MA stations). Microscopy observations
confirmed the presence of *R. intracellularis* at 5 SD stations of the MA and LD B and absence
at the SD stations and LD C of the SG. Noticeable was the co-occurence of free filaments of
*R. intracellularis* and degrading diatom cells (mainly belonging to the genus *Rhizosolenia*),
especially at the SD 5, 6 and 7.

12        The unicellular symbiotic groups, UCYN-A1 and A2 (and their respective hosts), were

the least detected targets. For example, UCYN-A1 was bd in 53% (63 of 120 samples) and
UCYN-A2 was bd in 66% (79 of 120 samples) of samples. UCYN-A1 and A2 represented <
0.4 % of total *nifH* genes detected and UCYN-A symbionts were bd in the SG, except at LD
C. When detected, average *nifH* abundance for UCYN-A1 and A2 were 8.60 x $10^4$ and 4.60 x
$10^4$ *nifH* copies L$^{-1}$, respectively, and usually accounted for <1.0-1.5 % of the total *nifH*
copies enumerated per station. One exception was at LD C in the SG, when UCYN-A1 and
A2 accounted for 4 and 11%, respectively, of the total *nifH* gene copies, and were the second
most abundant diazotroph (3.19 x $10^4$ and 8.53 x $10^4$ *nifH* copies L$^{-1}$). The average depth of
maximum *nifH* abundance for the UCYN-A1 and A2 symbionts was consistently recorded at
deeper depths (e.g. 55 and 58 m, respectively; 10 % light level).

23        The detection of the UCYN-A1 and A2 hosts mirrored the detection of their respective

symbionts. However, in 22 and 15 samples, respectively, the UCYN-A1 and A2 symbionts
were quantified while their hosts were bd. The UCYN-A hosts were never detected in samples





where their respective symbionts were bd or dnq. When both UCYN-A host and symbiont
were present, the abundances of the hosts were always one order of magnitude less than their
respective symbionts, with the exception of two samples for UCYN-A1 symbionts where their
respective host abundances were half, or nearly equal in abundance. UCYN-C was the least
abundant unicellular diazotroph and was only quantified in the 'at-sea' qPCR where detection
was poor and limited to the MA region (3 of 11 stations: 1-3 of 4 depths sampled) and
abundances never exceeded $10^2$ *nifH* copies L$^{-1}$ (Suppl. Table 3).
**3.3 Diazotroph and UCYN-A host covariation**
Several significant correlations between the target diazotrophs and hosts were identified (Fig.
3; Suppl. Table 4a). The *nifH* gene copy abundances of *Trichodesmium* and UCYN-B were
significantly positively correlated with each other ($p<0.01$). In addition, UCYN-B *nifH* gene
copy abundance was significantly positively correlated with those of both UCYN-A
symbionts (A1 and A2; $p< 0.01$) and UCYN-A2 host abundance ($p<0.05$). Abundances of
UCYN-A1 and A2 were significantly positively correlated with each other, and in addition,
with their respective host abundances ($p<0.01$). Lastly, the *nifH* copy abundances for het-1,
het-2 and het-3 were significantly positively correlated with one another, and with the *nifH*
copy abundances of *Trichodesmium* and UCYN-B ($p<0.01$). The only correlations that were
not significant were between the UCYN-A (including their hosts) and *Trichodesmium* and the
het-groups (with the exception of het-3, which correlated with the UCYN-A2 host ($p<0.05$)).
**3.4 Influence of environmental conditions on diazotroph and UCYN-A host abundances**
**in the WTSP**
The abundances of UCYN-A1 and A2 were significantly positively correlated with salinity
and depth ($p<0.02$ and $p<0.03$, respectively) (Fig. 3; Suppl. Table 4b). However, all other
diazotrophs were significantly negatively correlated with salinity and depth ($p<0.01$).
Moreover, *Trichodesmium*, UCYN-B, and the het-group (except het-3) were significantly



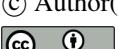

positively correlated (p<0.01) with PAR and temperature while UCYN-A1 and A2 were
significantly negatively correlated (p<0.02) with the latter parameters. All diazotrophic
targets, except UCYN-A1, UCYN-A2, and het-3, were significantly negatively correlated
(p<0.02) with DIN concentration. Similarly, all diazotrophs, except UCYN-A2, were
significantly negatively correlated (p<0.02) with DIP concentration, and all diazotrophs
except UCYN-A1, A2 and het-3 were significantly negatively correlated (p<0.001) with DiSi
concentration. The abundances of UCYN-A hosts, UCYN-A1 and UCYN-A2, and UCYN-B
were significantly correlated (p<0.001 and <0.05) with dissolved oxygen.  In general, the
correlations between abundances and several hydrographic parameters divided the diazotrophs
into two groups: the UCYN-A symbionts (and respective hosts) and all other diazotrophs.

11       Hierarchical clustering based on the Spearman's rank analyses resulted in the two major

groups: (1) a shallow and (2) deeper euphotic zone, inferred from the negative and positive
correlations, respectively, with depth (Fig. 3). For example, *Trichodesmium* and the symbiotic
het-1 and het-2 lineages characterize an upper water column group 1 with significant
clustering and positive correlations with temperature (p<0.001) and PAR (p<0.003), while
only UCYN-A1 and A2 symbionts and their respective hosts represent group 2. UCYN-B was
unique in an overlapping distribution, and resulted in positive significant correlations with
both the shallow (group 1) and deep (group 2) euphotic zone diazotrophs (e.g.
*Trichodesmium*, p<0.001 and UCYN-A1, p<0.004, respectively). The deeper dwelling group
2 significantly clustered and correlated positively with oxygen, depth, salinity and
fluorescence (p<0.03, except for UCYN-A2 and fluorescence, p<0.053). Despite clustering
with group 1, het-3 was less robust in a negative correlation with salinity (p<0.01).

23       The results from the Spearman's rank correlations were further confirmed and

visualized in the RDA biplot (Fig. 4a), which explains parameter importance (Fig. 4b).
Correlations with nutrients and PAR were omitted due to the limited number of data points.



Fluorescence, depth and salinity correlated positively with each other and negatively with
temperature, while oxygen was uncorrelated with all other environmental parameters. The
response variables UCYN-A1 and A2 and their respective hosts clustered with the
explanatory variables: fluorescence, salinity and depth, with a dependency towards oxygen.
On the other hand, the shallower euphotic group 1 (response variables *Trichodesmium*, het-1
and het-2) clustered closer to explanatory varaible temperature. In addition, most of the
observed variance is explained by the two axes RDA1 (72 %) and RDA2 (22 %), indicative of
depth and temperature, respectively, as the most important environmental parameters for
diazotroph abundance in our study. Together they form a depth-temperature gradient (RDA1)
where *Trichodesmium* occupies the warmest and shallowest waters, and UCYN-A occupies
the coldest and deepest waters, among the investigated cyanobacterial diazotrophs.
**3.5 Global drivers of diazotrophic abundance**
We found consistency between our results in the WTSP and the correlations identified in the
11 external datasets by the non-parametric correlation analyses and meta-analyses (Suppl.
Table 6). For example, in three of the external datasets, abundances of *Trichodesmium* spp.,
UCYN-B, and het-1, were significantly positively correlated with temperature and negatively
correlated with the same three parameters as in our study in the WTSP: salinity, DIP, and
DIN. The latter correlations were identified in two regions of the WTSP (tropical and
subtropical) and in the northern South China Sea (NSCS). In contrast to a significant positive
correlation between UCYN-A abundance and depth reported here in the WTSP, UCYN-A
abundance was negatively correlated with depth in 4 of the 11 external datasets (two regions
of the WTSP, Tropical Atlantic (TA), and NSCS). Moreover, and consistent with several of
the other diazotrophs (*Trichodesmium*, UCYN-B, het-1), UCYN-A abundance was negatively
correlated with DIP and DIN concentrations (5 and 3 additional external datasets,
respectively) (Suppl. Table 6).




Meta-analysis revealed similar groupings (e.g. shallow and deep) as observed in the
WTSP, however, the significance was less robust (Suppl. Table 6). For example abundances
of *Trichodesmium* and het-1 and het-2 were significantly positively correlated with
temperature and negatively correlated with salinity ($p < 0.05$).  No significance was found for
UCYN-A abundance for the latter parameters, and UCYN-B abundance was un-correlated
with salinity and significantly positively correlated with temperature ($p < 0.05$). In addition,
UCYN-A was the only diazotroph that was uncorrelated with het-2, while all other
diazotrophs had a significant positive correlation with het-2 ($p < 0.05$). Similar to our findings
reported for the WTSP, all diazotrophs, except UCYN-A, correlated significantly negatively
with depth, DIP and DIN concentrations ($p < 0.05$) (except het-2 with DIP which was not
significant). Finally, UCYN-B and het-1 abundances were significantly negatively correlated
with chl *a* ($p < 0.05$), while *Trichodesmium*, UCYN-A and het-2 were uncorrelated.
**3.6 Cross reactivity tests**
No amplification was detected for the newly designed UCYN-A1 host oligonucleotides run
with the UCYN-A2 as template DNA and vice versa (Suppl. Fig. 1a).
Running the het assay in fast mode showed a lower cross-reactivity between the het-1 assay
and the het-2 template than vice versa (the het-2 assay and het-1 template) (Suppl. Fig. 1b). In
fact, no amplification was detected in the last two template additions and the Ct differences
were > 9 when het-1 assay was run with het-2 templates. The UCYNA-2 assay detected the
UCYN-A1 template in all but the last template addition and with Ct differences >3 (1 order of
magnitude) while there was a 18-20 difference in Ct value (less gene copies) when UCYN-A1
assay was run in fast mode with UCYN-A2 templates at either annealing temperature (60° C
or 64° C) and only the first three template additions ($10^8$-$10^6$ *nifH* copies µL$^{-1}$) were detected
(Suppl. Fig. 1c-d).





**4 Discussion**
**4.1 Environmental conditions in the WTSP**
The SP is one of the most oligotrophic regions of the world's oceans with chronically low
dissolved nutrient concentrations, especially DIN, and thus, is considered an area primed for
$N_2$ fixation. Likewise, we encountered surface hydrographic conditions in the WTSP that
were consistently low in dissolved nutrient concentrations and similar to earlier reports for the
equatorial Pacific (Bonnet et al., 2009; Dufour et al., 1999; Moutin et al., 2008; Van Den
Broeck et al., 2004). The conservative tracers of temperature and salinity remained constant in
the surface between the MA and SG regions, hence the elevated nutrient concentrations in the
SG is likely not related to an eddy intrusion. The deviation away from a 16:1 relationship
(Redfield ratio) (data not shown) in the upper 125 m in both regions (MA and SG) was
indicative of DIN limitation. The low DIP concentrations in MA waters suggest utilization of
DIP by diazotrophs in the absence of DIN, and likely other sources of nitrogen were available,
e.g. dissolved organic nitrogen or $N_2$ fixation (Karl et al., 2001).
**4.2 Detection of diazotrophs and application of 'at sea' qPCR**
*Trichodesmium,* UCYN-B, and the het-groups are easily identifiable by standard epi-
fluorescence microscopy, and so these populations can readily be observed 'at sea'. However,
the UCYN-A1 and UCYN-A2, and their respective hosts, require a lengthy fluorescent *in situ*
hybridization (FISH) protocol that is difficult to implement in the field. On the other hand,
nowadays oceanographers have a suite of other molecular genetic tools, some of which are
also 'sea-going' and autonomous (e.g. Robidart et al. 2014; Ottesen et al. 2013; Preston et al.
2011), thereby making quantification of microscopically unidentified microorganisms
tangible by quantifying their genes, simultaneous with collection of hydrographic data. Here,
we showed a rather efficient, steadfast (within 3 hrs of sample collection), and 'sea-going'
nucleic acid extraction and qPCR to quantify diazotrophs by their *nifH* gene, which was used



in real time during the OUTPACE cruise to help locate the LD stations for the purpose of the
project (see Moutin et al., this issue). The comparisons of the 'at sea' assays to the lab-based
full extraction protocol and qPCR on archived samples indicated that the assays were
consistent, and surprisingly the shortened DNA extraction performed 'at sea' had higher
abundances for all three targets (UCYN-A1, UCYN-A2 and UCYN-B) in 16-25 % of the
samples processed, depending on the target diazotroph. The 'at sea' (and lab-based) qPCRs
could be appended with a multi-plexing approach to both increase and broaden the number of
metabolic pathways (e.g. *narB, rbcL, nirS*) and/or phylotypes quantified simultaneously.
**4.3 Abundance and vertical distribution of diazotrophs in the WTSP**
Earlier work based on N isotope ratios (delta $^{15}$N) of suspended particulate matter and
dissolved organic N (DON) in the WTSP suggested that new production is likely fueled by $N_2$
fixation in this region (Hansell and Feely, 2000; Yoshikawa et al., 2005). The SP is also an
area where high abundances of the unicellular diazotrophs, in particular UCYN-A and
UCYN-B, have been previously reported (Biegala and Raimbault, 2008; Bonnet et al., 2009,
2015; Moisander et al., 2010) and account for a significant (74%) portion of the areal $N_2$
fixation (Bonnet et al., 2009). Hence, it was likely to encounter the presence of diazotrophic
populations.

18        Recently UCYN-A and its various lineages have been highlighted as one of the most

widespread and abundant diazotrophs (Farnelid et al., 2016 and references therein), which has
led to the dramatic shift in the canonical paradigm of *Trichodesmium* as the only significant
diazotroph. Surprisingly, here, we report abundances of the UCYN-A1 and UCYN-A2
lineages that are comparatively lower than earlier reports. In fact, UCYN-A1 and A2 were the
least detected diazotrophs. Both UCYN-A phylotypes were largely restricted to the MA, with
the exception of high densities ($3.2 \times 10^4$ and $8.5 \times 10^4$ *nifH* copies $L^{-1}$, respectively) found at
one depth (60 m) of LD C, which borders the MA region. Consistent with higher UCYN-A



biomass at depth at LDC were microscopy observations of high abundances of picoeukaryotes
similar in size and shape previously reported for the UCYN-A hosts (Krupke et al. 2013).
The vertical distribution of UCYN-A1 (and A2) was similar to Moisander's et al. (2010) and
others, including earlier studies in the North Pacific Ocean (NP) and NA, where maximum
abundances of UCYN-A are common to deeper depths in the euphotic zone (below 45 m)
(e.g. Bonnet et al., 2015; Foster et al., 2007; Goebel et al., 2010; Needoba et al., 2007).
Likewise, we also observed as others (Cabello et al., 2016) that the UCYN-A based
symbioses co-occur and typically have decreased abundance towards the DCM, and
maximum abundances slightly above the nitracline.
Unlike UCYN-A phylotypes, UCYN-B and *Trichodesmium* were the most abundant
diazotrophs in the WTSP, and UCYN-B in particular was the most detected phylotype (99%
detection; dnq or higher in 119 of 120 samples). High abundances of *Trichodesmium* in the
upper 10 m, including presence of surface slicks and free filaments, was widespread in the
MA region and consistent with earlier observations of high surface densities further north in
the SP (Moisander et al., 2010; Shiozaki et al., 2014). Surface slicks have also been reported
elsewhere, e.g. the North Atlantic (NA) (Goebel et al., 2010; Langlois et al., 2005). The depth
of maximum abundance for *Trichodesmium* deepened from the MA (10 m) region to the open
gyre (SG, 31 m), which was similar to earlier reports in the equatorial Pacific (Bonnet et al.,
2009). A niche partitioning has been suggested for *Trichodesmium* and unicellular diazotrophs
in the SP (Bonnet et al., 2015; Moisander et al., 2010) and elsewhere (Goebel et al., 2010;
Langlois et al., 2005; Messer et al., 2015). However, here in the WTSP, *Trichodesmium*
abundance was correlated with UCYN-B, which is consistent with previous studies in other
ocean basins, e.g. Atlantic Ocean (Foster et al., 2007, 2009; Langlois et al., 2008), and the
South China Sea (Moisander et al., 2008). UCYN-B co-occurred with *Trichodesmium* in the
surface samples, although at lesser *nifH* copy abundances, and more often UCYN-B had



subsurface maxima (35-70 m) in both regions (MA and SG) of the transect. The latter is also
consistent with Moisander et al. (2010) who observed maximum abundances of UCYN-B
north of the Fijian islands at 37m.
All 3 heterocystous symbiont phylotypes co-occurred and were widespread in the MA,
with het-1 as the most abundant and most highly detected het group (70% detection or 84 of
120 samples). The early work of Moisander et al. (2010) detected het-1 in all but one of 26
stations sampled (56% detected, or 56 of 100 samples), and highest *nifH* copy densities were
reported north east of our cruise transect. Moreover, Bonnet et al. (2015) detected het-1 and
het-2 at the surface of one out of 10 stations west (approximately 10 degrees W) of our cruise
transect. Het-2 and het-3 were not quantified by Moisander et al. (2010) and het-3 was not
quantified by Bonnet et al. (2015). Therefore our study is among the first to report on the
abundances and distributions for all 3 heterocystous diazotrophs in a large expanse of the SP.
The 3 het phylotypes were however recently reported from a mesocosm (enclosed design)
experiment in the Noumea lagoon, a low nutrient low chlorophyll (LNLC) region located
along the New Caledonian coast (Turk-Kubo et al., 2015). In fact, het-1 and het-2 were among
the most abundant diazotrophs in the first half of the experiment (Turk-Kubo et al., 2015).
Two additional earlier studies have also reported microscopic observations of free-living
*Richelia* in the same lagoon (Biegala and Raimbault, 2008; Garcia et al., 2007).
Highest densities ($10^4$-$10^6$ *nifH* copies L$^{-1}$) of the *Richelia* phylotypes were restricted to
the western region of the MA, and in the upper 12 m, which is shallower than the subsurface
maximum (25 m) commonly reported for het-1 (and het-2) in the Western Tropical North
Atlantic (WTNA) and NP (Church et al., 2005; Foster et al., 2007; Goebel et al., 2010). Our
microscopy observations from SD 5-7 and LD A indicated that near surface *Rhizosolenia*
populations were in a moribund state since frustules were broken and free filaments of
*Richelia* were observed. Our observations also coincide with a region of high backscattering



measurements in the upper water column (5-30 m) (Dupouy et al., this issue). Het-1 *nifH*
copies were 4 orders of magnitude higher in abundance in the moored sediment traps of LD A
(325 m: 2.0 x $10^7$ *nifH* copies L$^{-1}$) and LD B (325 and 500m: 5.8 x $10^6$ and 1.10 x $10^7$ *nifH*
copies L$^{-1}$, respectively) (Caffin et al., this issue) than the *nifH* copies detected in the
overlying waters (3.11 x $10^3$ *nifH* copies L$^{-1}$ and 4.1 x $10^2$ *nifH* copies L$^{-1}$, respectively).
Combined, the latter observations suggest that a higher density of the het-1 population was
likely present prior to our sampling and perhaps derived from a 'seed' population originating
in the coastal regions of New Caledonia.

9       The UCYN-C phylotype was poorly detected in the 'at sea' assays (61% samples were

bd and maximum abundance was 5.0x$10^2$ *nifH* copies L$^{-1}$), and as such was not enumerated in
the archived samples. The low detection of UCYN-C is consistent with Taniuchi et al. (2012),
who estimated that UCYN-C only represented a small portion of diazotrophs detected in the
NW Pacific (Kuroshio Current). However, a recent study reported relatively high UCYN-C
abundances in the open waters of the Solomon Sea (north of the MA) (Berthelot et al.,
submitted). UCYN-C has also been observed in the New Caledonian lagoon (Turk-Kubo et
al., 2015), where it was the most dominant diazotroph in the first part of the aforementioned
mesocom experiment (Turk-Kubo et al., 2015).  Like most plankton, abundances can be
patchy as was observed with UCYN-C in our study.
**4.4 UCYN-A and host (co)-occurrence**
Earlier and recent work has suggested a high host dependency (e.g. smaller and streamlined
genomes), and selectivity in the UCYN-A based symbioses (Cabello et al., 2016; Cornejo-
Castillo et al., 2016; Farnelid et al., 2016; Krupke et al., 2013, 2014; Thompson et al., 2012;
Tripp et al., 2010). Moreover, the UCYN-A partnerships are also considered mutualistic,
where the host and symbiont both benefit by exchange of metabolites (e.g. reduced C and N,
respectively) (Krupke et al., 2014; Thompson et al., 2012); hence one would expect parallel





distributions for both partners. Some have argued that the partnership is also obligatory since
few observations of free-living hosts have been reported and abundances of free symbionts
assumed to be derived from disruption during sample preparation are always correlated with
their hosts (Cabello et al., 2016; Krupke et al., 2014; Thompson et al., 2012). Thus, by use of
our newly designed oligonucleotides for the UCYN-A1 host and previously designed
oligonucleotides for the UCYN-A2 host (Thompson et al., 2014), we unexpectedly found that
both UCYN-A1 and A2 were often (89% and 59%, respectively; not considering dnq)
detected in the absence (or bd) of their respective hosts, while the hosts, when detected,
always coincided with increased UCYN-A abundance. Our observations could result if the
UCYN-A lineages can live freely, or in either a loose association, or perhaps with a wider
range of hosts than previously thought and detected by the UCYN-A host assays. Presence of
UCYN-A in the absence of their respective hosts could also indicate that the growth of
symbiont and host is asynchronous, a pattern reported once in the het-1 or *Rhizosolenia-*
*Richelia* symbioses (Villareal 1989).
The number of cells per partner lineage is considered specific as well, such that 1-2
UCYN-A1 cell is associated with a prymnesiophyte partner (UCYN-A1 host) and the larger
*B. bigelowii* (UCYN-A2 host) host associates with multiple and variable numbers of UCYN-
A2 cells to compensate for its higher N requirement (Cornejo-Castillo et al., 2016). On the
contrary, we found evidence that there are multiple UCYN-A1 and A2 symbionts in both host
types, which is somewhat surprising given that the host target gene (18S rRNA) is a multiple
copy gene, meaning that we would expect higher gene copy numbers for each host.
Nonetheless, we consistently observed higher abundances for the UCYN-A1 and A2
symbionts than their respective hosts. UCYN-A1 and A2 were 2-10 and 6-34 times,
respectively, more abundant than their hosts. A symbiosome-like compartment has also been
described attached to the UCYN-A2 host or residing free (Cornejo-Castillo et al., 2016).



Thus, one plausible explanation for the higher abundances of the UCYN-A2, in particular, in
the absence of their respective host, could result if our assays quantified UCYN-A2 residing
in a dislodged free-floating symbiosome, or an overestimate of the UCYN-A2 due to cross-
reactivity with UCYN-A3 lineage as expected by *in silico* tests (Farnelid et al. 2016). It is
less likely that the UCYN-A2 was overquantified due to cross-reaction with UCYN-A1
templates since our cross-reactivity tests showed a weak cross reaction (see below).
**4.5 Environmental influence on diazotroph abundances and distributions**
The annual N inputs through biological $N_2$ fixation in the oceans is considered high, ranging
100-200 Tg N (Eugster and Gruber, 2012; Luo et al., 2012), yet large uncertainties remain in
what factor(s) influence the abundance, distribution, and activity of marine diazotrophs.
Initially, we hypothesized that conditions favoring a particular cyanobacterial diazotroph
would differ given the contrasting life histories (free-living, colonial, and symbiotic).
Moreover, we also suspected that the conditions promoting DDAs would differ from those
influencing the UCYN-A based symbioses given the vast differences in the symbionts and
hosts (e.g. genome content of symbiont, cell size of symbiont and hosts in the two systems;
expected number of symbionts/host; host phylogeny: diatom vs. prymnesiophyte). Thus,
determining the condition or sets of conditions that drive cyanobacterial diazotroph
distribution, abundance, and activity is of great interest.

19       Hydrographic conditions and dissolved nutrient concentrations measured at the time of

sampling were used to correlate diazotrophic abundance with various environmental
parameters. Consistently, in two independent statistical tests, two groups emerged in the
WTSP: 1) UCYN-A1 and A2 and their respective hosts 2) het-1, het-2 and het-3, UCYN-B
and *Trichodesmium*. Thus, agreeing with our initial hypothesis that conditions favoring the
UCYN-A based symbioses does differ from the conditions for DDAs, and in addition for the
free-living cyanobacterial diazotrophs.



Temperature is often cited as the most important driver of diazotroph abundance and
distribution (Messer et al., 2016; Moisander et al., 2010). As shown earlier in the WTSP, both
*Trichodesmium* spp. and UCYN-B were most abundant in warmer surface waters (> 27 ℃) in
the north, while UCYN-A dominated in the cooler (24-26 ℃) southern waters of WTSP
(Bonnet et al., 2015; Moisander et al., 2010). Likewise, we found similar abundances and
temperature optima for the latter three diazotrophs and significant correlations between the
various diazotrophs and temperature. In fact, all diazotrophs, except the UCYN-A lineages
were significantly positively correlated with temperature in the WTSP. In addition to
temperature, environmental parameters PAR, salinity and depth were also significantly
influencing abundance and distribution. Moreover, the latter two variables drove the
abundances of UCYN-A symbioses (A1 and A2) apart from the rest of the diazotrophs in the
WTSP, including both free-living phylotypes and the symbiotic heterocystous lineages.
The maximum abundances at depth for UCYN-A1 and UCYN-A2 were slightly above
or at the nitracline and coincided with higher measures of fluorescence from the CTD. The
latter is consistent with observations of high UCYN-A abundances in coastal habitats
(Bombar et al., 2014), estuaries (Messer et al., 2015), or in waters that are recently entrained
with new nutrients (Moisander et al., 2010). Increased *nifH* copies and/or *nifH* gene
expression for UCYN-A have also been reported from bioassay experiments amended with
nutrients, including DIN, phosphate and iron (Krupke et al., 2015; Langlois et al., 2012;
Moisander et al., 2012). The latter is in contrast with the data reported here in the WTSP
(including the meta-analysis) and several of the external datasets (e.g. WTSP, TA, NA,
NSCS), which finds a negative correlation between DIN and DIP concentrations and
abundance of most of the diazotrophs, including UCYN-A. In the WTNA, waters with high
DiSi concentration and low N:P ratios, driven by a disproportionate utilization of N relative to
P, results in consistent and widespread blooms of the *Hemiaulus-Richelia* symbioses (het-2)





(Foster et al., 2007; Subramaniam et al., 2008). Across the cruise transect, DIP and DiSi
concentrations were considered not limiting (Thierry Moutin, this issue), while DIN was
below detection, hence conditions favoring symbiotic diatoms, and as reported here, the
higher abundances of het-1 *nifH* gene copies and observations of *Rhizosolenia* hosts in the
MA.

6        All the diazotrophs described here are either photoautotrophic or associated with

photoautotrophic partners (UCYN-A, het-group). Therefore, light irradiance (e.g PAR) and
availability will impact the abundance and distribution of the diazotrophic populations.
Moreover, and related to light availability is the influence of day length or changes in the
photoperiod which can influence diazotroph distribution, in particular the symbiotic diatoms
(Karl et al., 2012). Results from CARD-FISH observations of the UYCN-A1 and A2
symbioses have reported a strong dependency on light intensity, which results in higher
abundances nearer to the surface (Cabello et al., 2016). Presence in shallower waters is also
thought to be strategic for avoiding competition (Cabello et al. 2016). However, in the WTSP,
in 11 of the 14 stations where UCYN-A1 and A2 were detected at sub-surface depth maxima,
the same lineages (and corresponding hosts) were undetected at the surface and a negative
correlation was found with PAR. Microscopy observations also confirmed higher numbers of
pico-eukaryotes at depth. Hence, it would appear that low light was a pre-requisite for high
abundances of UCYN-A; while the other free-living diazotrophs and symbiotic het-1 and het-
2 were positively correlated with PAR, and had maxima closer to the surface with higher
PAR. Interestingly and unexpected was the lack of correlation between PAR and the UCYN-
A host lineages, especially since it is the host partners that require light for photosynthesis.

23        In an attempt to identify the consistency in the correlation patterns identified in the

WTSP with other regions of the world's ocean, the same statistical analyses were performed
on 11 publically available datasets and subsequently run through a meta-analysis. Our



statistical analyses provided coefficients and p-values for easy evaluation and comparisons
between data sets for the influence of environmental parameter(s) and diazotrophs abundance.
It confirmed that UCYN-A indeed stands out from the other diazotrophs in terms of
environmental parameter influence, mainly by being uncorrelated with temperature, which for
all other diazotrophs was a positive correlation. For most other environmental variables the
pattern for UCYN-A does not hold true in the meta-analysis. However, for the other
diazotrophs depth and salinity follow the same pattern as observed in the WTSP (except for
UCYN-B being uncorrelated with salinity). Furthermore, what did unify all diazotrophs in the
meta-analysis were their consistent negative correlations between abundance and
concentrations of DIP and DIN, which was also observed in the WTSP and again UCYN-A
was the exception.

12       In summary, the correlations observed in the WTSP were not always consistent with the

meta-analysis of the external datasets. We attribute the inconsistencies in part to seasonal
differences in sample collections, and the impact of an individual environmental parameter or
sets of parameters on a local and regional scale that make it difficult to unambiguously
explain the abundance and distribution patterns. Unlike our initial hypotheses, determining the
condition or sets of conditions favoring one diazotroph or life history strategy (free-living vs.
symbiotic) is complex and likely not all diazotrophs are influenced by the same condition in
time and space.
**4.6 Estimation of diazotrophs by nifH qPCR**
When interpreting abundance estimates by qPCR there are a few assumptions to keep in mind.
A caveat of qPCR assays assumes that there is one gene copy per cell.  However, recent
evidence in filamentous and heterocystous cyanobacteria reports evidence of polyploidy
dependent on cell cycle (Griese et al., 2011; Sargent et al., 2016; Sukenik et al., 2012).
Moreover, *Trichodesmium* may contain up to 100 genome copies per cell (Sargent et al.,



2016), thus a potential for overestimation. On the other hand, underestimation by qPCR is
also plausible if one considers that DNA extraction efficiency is not 100% and can vary
between species and DNA extraction kits (Mumy and Findlay, 2004), and if high probe
specificity favors exclusion of closely related phylotypes for a particular target or lineage.

5        A final consideration with qPCR as shown here, is the degree of cross-reactivity in

assays targeting closely related lineages (e.g. UCYN-A and het). Oligonucleotide specificity
as a source of underestimation of the UCYN-A lineages was recently reviewed by a *de nova*
analyses (Farnelid et al., 2016) showing the potential to underestimate UCYN-A sublineages
since the widely used oligonucleotides for UCYN-A1 contains several mismatches to the
other UCYN-A sublineages. The latter becomes important when the sublineages co-occur.
Here, however, we highlight the potential to overestimate. For example, UCYN-A2
oligonucleotides amplified the UCYN-A1 templates, indicating a tendency to overquantify
UCYN-A2 in the presence of A1. Moreover, when the annealing temperature was set to 64
℃, to distinguish between UCYN-A1 and A2 as recommended by Thompson et al. (2014),
the assay still failed to separate the two sub-lineages when run in fast mode. Thus, the fast
mode feature has a shortcoming that could influence a wider range of targets than the ones
presented here. We observed the same cross-reactivity reported earlier (Foster et al., 2007) for
het-1 and het-2 when run in fast mode and highlights the potential to overestimate het-2 if het-
1 co-occurs at densities approximately $10^6$ *nifH* copies L$^{-1}$. The latter observation has never
been reported.
**Conclusions**

22       Consistent with earlier observations in the WTSP, we found diazotrophic cyanobacteria

to be abundant. The most abundant cyanobacterial diazotrophs were UCYN-B,
*Trichodesmium* and the symbiotic *Richelia* lineage het-1. Although the cell integrity and
detection of het-1 in water column samples and those from depth (e.g. sediment traps)



indicated that the populations were in a senescent state, our work represents one of the first
documentation of the three DDA populations in a wide expanse of the WTSP. In contrast to
earlier work in the SP and other recent reports from global ocean surveys (Farnelid et al.,
2016; Martínez-Pérez et al., 2016), we observed low abundances and poor detection of both
UCYN-A (A1 and A2) lineages. According to our qPCR results, UCYN-A was also
enumerated when their respective hosts were below detection, which contrasts to the assumed
high fidelity and dependency in the partnerships; however, we cannot discount that the
disparity in host-symbiont detection was not a result from qPCR oligonucleotide assay bias
and/or overestimations indicated by our cross-reactivity tests.
Our initial hypothesis was that the condition or sets of conditions, which promote the
distribution of one diazotroph, would differ. Moreover, the parameters for symbiotic
diazotrophs should also differ from that of free-living phylotypes, and given the vast
difference in hosts (diatoms and prymnesiophyte, respectively) and genome content for the het
and UCYN-A symbionts, we further hypothesized divergent conditions favoring one
symbiosis over another. In the WTSP, the same conditions favored abundances of both the
free-living phylotypes and the diatom (het groups) symbioses. However, the same conditions
impacted the abundance of UCYN-A based symbiosis negatively, hence, somewhat
supporting our intial hypothesis that conditions for one symbiosis type would differ. In the
external datasets, however, we observed differences in environmental conditions favoring
abundances of the investigated diazotrophs compared to the WTSP, which underscores that
diazotrophs are not similarly influenced by the same condition in time and space.
Multivariate approaches on numerous parameters and with high spatial resolution are
required to understand the complex and often indirect effects that govern species distribution.
Finally, this study highlights reliable quantification of *nifH* genes for various $N_2$ fixing
cyanobacteria 'at sea' in the tropical open ocean and how environmental parameters influence



distribution and abundance of diazotrophs differently both regionally and across ocean basins.
However, it is of great interest to know, if the same parameters influence gene expressions
(e.g. *nifH*), and ultimately $N_2$ fixation rates, in the same manner, thus, understanding the
weight of environmental parameters influencing diazotrophic abundance and distribution.
Given the global significance of $N_2$ fixation as a major new source of N to the oceans, the
metanalysis presented here could be directly applicable to improving parameter constraints on
model-based approaches for predicting areas prone to diaztrophy.



## 1 Competing interests

The authors declare that they have no conflict of interest.

## 3 Acknowledgements

The participation (MS and AC), sample processing, and work presented here is supported by Knut and Alice Wallenberg foundation (to RAF). This is a contribution to the OUTPACE (Oligotrophy from Ultra-oligoTrophy PACific Experiment) project funded by the French research national agency (ANR-14-CE01-0007-01), the LEFE-CyBER program (CNRS-INSU), the GOPS program (IRD) and the CNES (BC T23, ZBC 4500048836). The OUTPACE cruise (http://dx.doi.org/10.17600/15000900) was managed by T. Moutin and S. Bonnet from the MIO (Mediterranean Institute of Oceanography). T. Moutin is specifically thanked for invaluable input and feedback on this paper. The authors thank the crew of the R/V *L'Atalante* for outstanding shipboard operation. G. Rougier and M. Picheral are thanked for their efficient help in CTD rosette management and data processing, as is Catherine Schmechtig for the LEFE CYBER database management. All data and metadata are available at the following web address: http://www.obs-vlfr.fr/proof/php/outpace/outpace.php. Special acknowledgement to Andreas Krupke (formerly of WHOI), Kyle Frischkorn (WHOI), Mar Benavides (MIO/IRD), Hugo Berthelot (MIO), and Mathieu Caffin (MIO). Olivier Grosso (MIO) and Sandra Hélias (MIO) are acknowledged for nutrient analyses. Additionally we thank Dr. Lotta Berntzon for assisting in sample processing and Konrad Karlsson for assistance with the multivariate statistics.



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



1    Table 01. Summary of environmental conditions in the surface along the cruise transect.

| Region | Stations | surface* DIN‡ µM | surface* DIP µM | surface* DiSi µM | surface* salinity (PSU) | surface* temp. ˚C |
|---|---|---|---|---|---|---|
| Melanesian archipelago (MA) 160˚ E-178˚ E 170 ˚W - 175 ˚W | SD1-12 LDA LDB | 0.02 ± 0.01 | 0.03 ± 0.02 | 0.55 ± 0.10 | 35.13 ± 0.27 | 29.33 ± 0.45 |
| Subtropical gyre (SG) 160 ˚W- 169˚W | SD13-15 LDC | 0.01 ± 0.01 | 0.18 ± 0.07 | 0.79 ± 0.04 | 35.12 ± 0.10 | 29.34 ± 0.18 |

2    *5m depth, ‡NO$_2$+NO





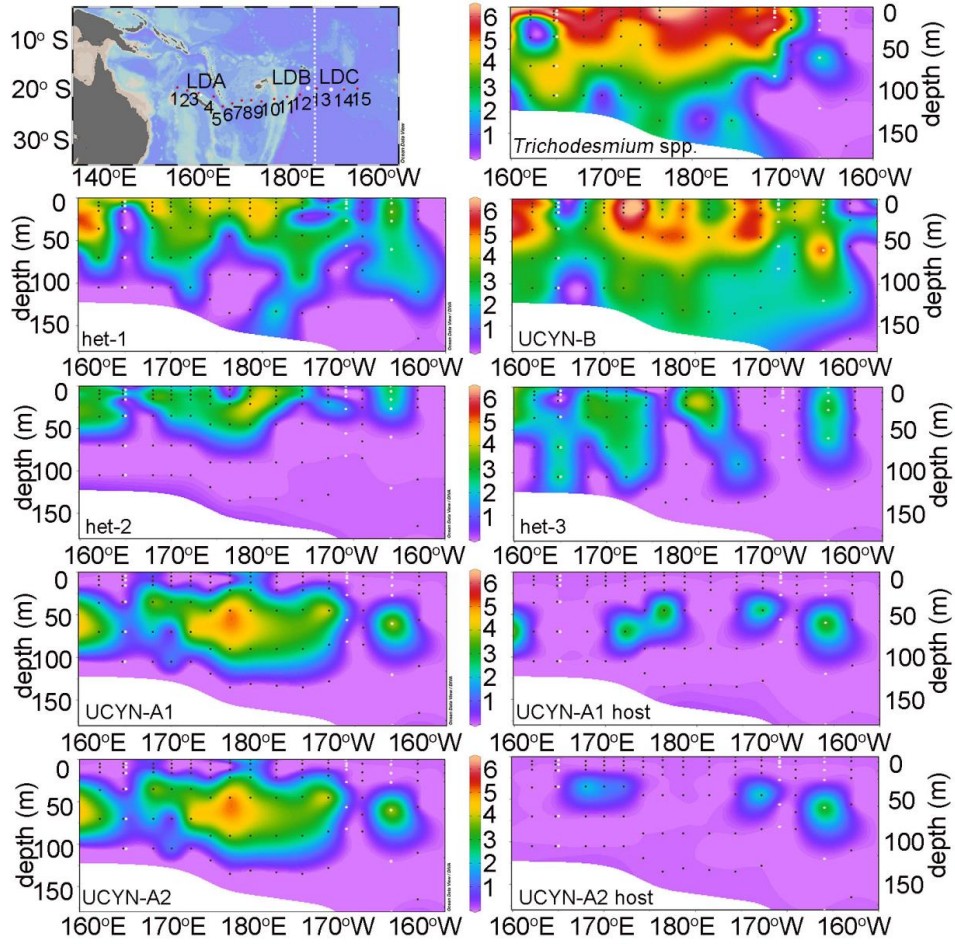

**Figure 1.** Sampling locations and the horizontal and vertical distributions of diazotrophs and

the UCYN-A1 and UCYNA-2 hosts in the study area. Sampling depths are indicated as black

dots (white for LD stations) and the abundances are the log *nifH* gene copy L$^{-1}$ for the

diazotrophs and 18S rRNA gene copies L$^{-1}$ for the UCYN-A host lineages.





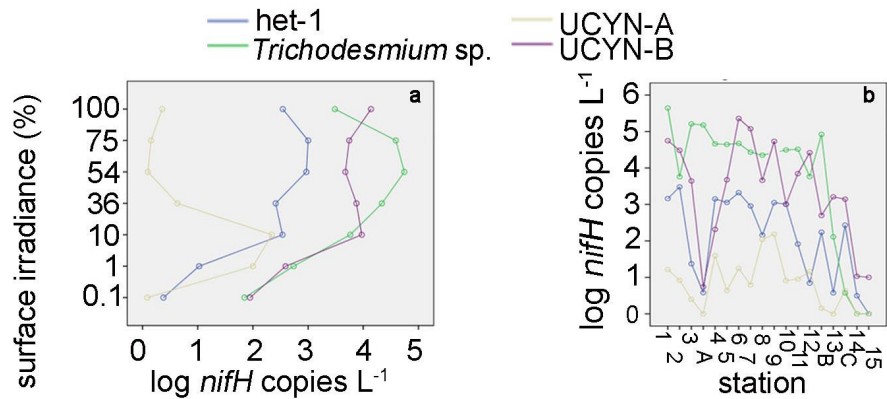

**Figure 2 a-b.** LOG10 transformed mean abundances for the 4 most abundant diazotrophs
across the transect: het-1 (blue), *Trichodesmium* (green), UCYN-A (yellow) and UCYN-B
(red). The mean *nifH* abundance values (log n*ifH* copies L$^{-1}$) shown as a function of (a)
percent (%) surface irradiance and (b) at each station





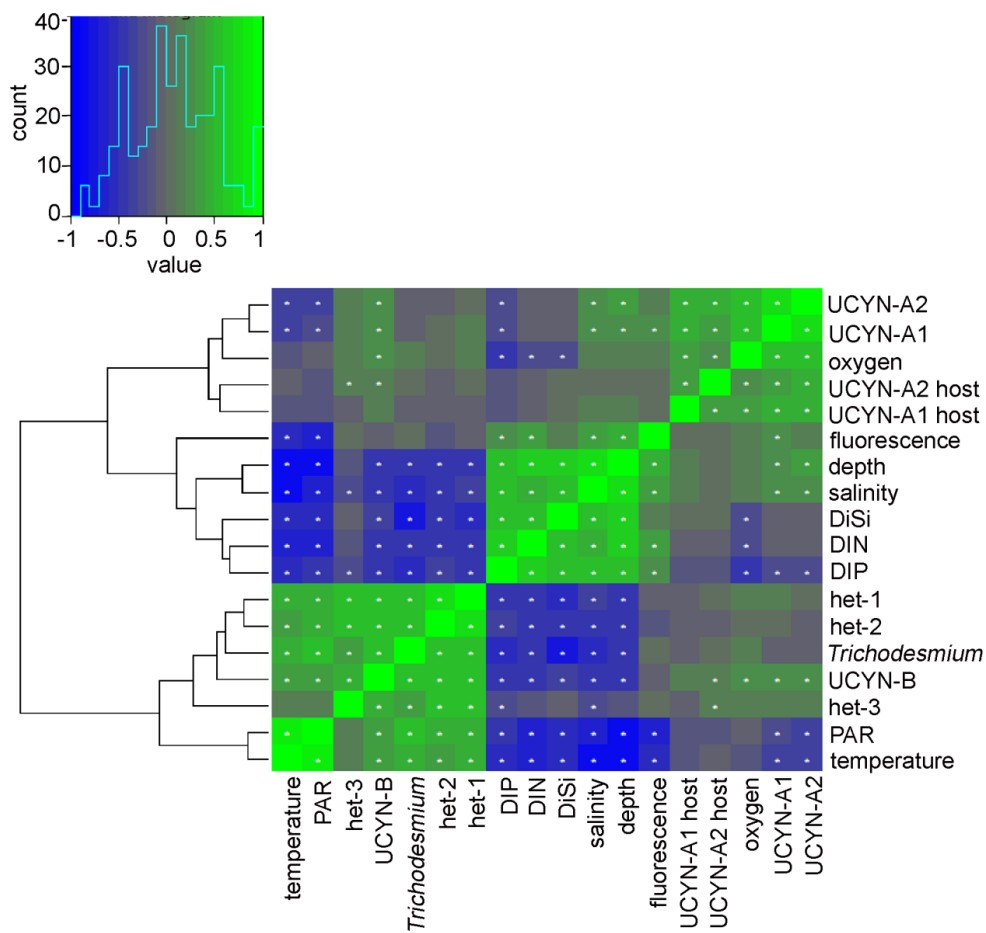

2    **Figure 3.** Hierarchical clustering heat map of Spearman's Rho results. The histogram shows

3    negative (blue) and positive (green) values of correlation strength between parameters. Stars

4    within cells mark significant correlations (p<0.05).





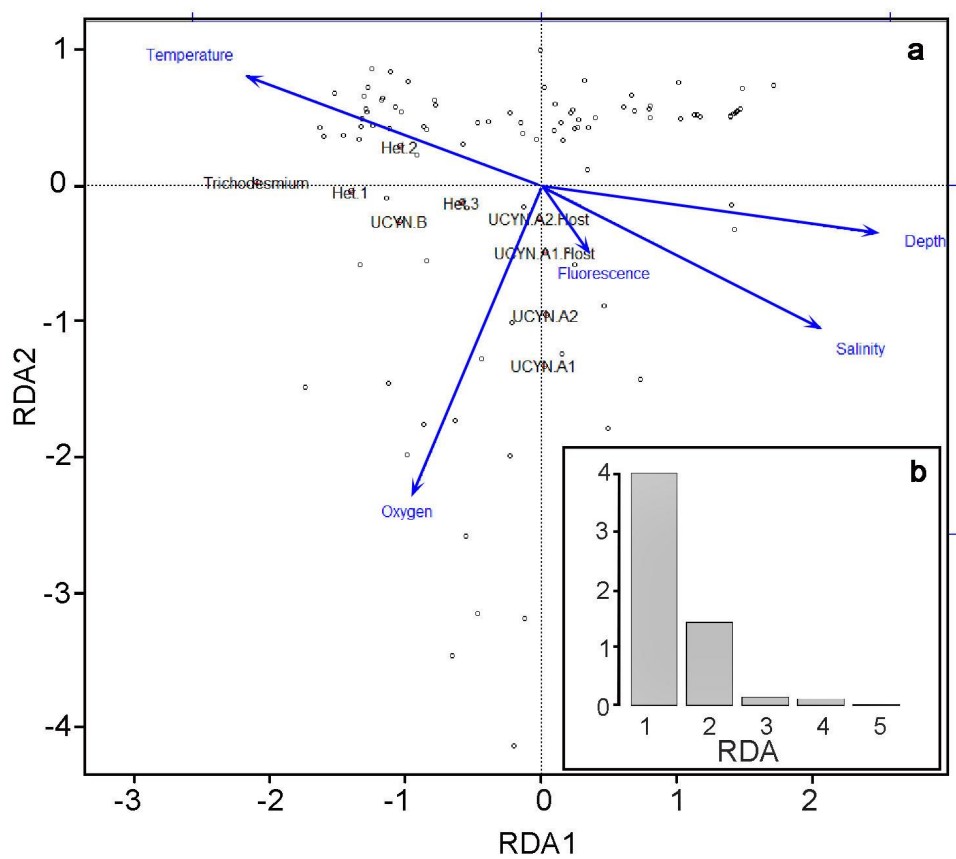

**Figure 4 a-b.** Multivariate RDA biplot (a), which also depicts variance of included

parameters (b). As can be seen, a majority of the variance in the dataset is explained by the

RDA1 and RDA2 axes meaning that most of the variance observed is explained by the

included environmental parameters. The arrows are the constrained explanatory vectors with

the dots representing the superimposed unconstrained response variables. PAR and nutrients

(DIP and DIN) were omitted due to limited data.