# Peer review of "Distribution and drivers of symbiotic and free-living"

_Biogeosciences, 2017_

## Referee Comment (RC1) · Anonymous Referee #1 · 23 Apr 2017

**General Comments**

Stenegren et al. utilise *nifH* specific quantitative PCR (qPCR) to determine the abundances of a suite of cyanobacterial diazotrophs in the western tropical South Pacific Ocean. The author's present results from both 'at sea' and laboratory based qPCR assays for unicellular (UCYN) groups, and identified interesting differences in absolute quantification between the two. In addition, Stenegren et al. present a new qPCR assay to quantify the picoeukaryote host of UCYN-A1. Correlation analyses were used to determine environmental constraints on the abundances of the different diazotroph groups, and include a meta-analysis of other publicly available datasets to expand their findings to other ocean regions. This work will be of interest to the readership of Biogeosciences as it reports on diazotroph abundances in a relatively understudied marine province, which has recently been identified as a potential hotspot for biological $N_2$ fixation.

I am surprised that only the picoeukaryote hosts of UCYN-A were quantified by qPCR, but the hosts of the heterocystous diazotrophs were not, this could have strengthened the authors ability to address their hypotheses and test the underlying environmental factors influencing the different symbioses.

I think the discrepancies between the 'at sea' and lab based qPCR UCYN targets should be discussed further, some of the differences in *nifH* copies are quite large and this could have implications for other studies that only perform qPCR on DNA samples post-sampling, or vice versa. Could the two different DNA extraction methods that were used cause these differences? Were the samples that are being compared taken from the same Niskin bottles, or could the differences shown in Supplementary Table 2 be due to natural heterogeneity in microbial communities sampled at slightly different times while on station?

In general, I think that the manuscript would benefit from some re-structuring of the Introduction and the Results prior to publication, and could be refined to streamline the main purpose of the study and to highlight the main findings.

**Specific comments**

Abstract:

In general, the Abstract could be improved by including the quantitative *nifH* copy numbers rather than the percentage of *nifH* copies detected.

Line 9. What does the >47 % refer to when you say the UCYN-A lineages were poorly detected?

Lines 9-12. This is inconsistent- the hosts mirrored their respective symbionts yet were below detection?

Lines 14-15. Perhaps re-phrase to state that they correlated with the surface group. Include the parameters that were significantly correlated with the deep UCYN-A group too.

Lines 15-16. Could you expand on this briefly?

Line 18. 'free-living cyanobacterial diazotrophs'

Introduction:

Pg 4 lines 18-19. There is no clear link between these two paragraphs. Suggest moving the next paragraph, from page 5 lines 14-23, up to provide the link between the diversity of cyanobacterial diazotrophs and the introduction to the environmental characteristics of the WTSP.

Pg 5 lines 18-23. Perhaps expand on this slightly to indicate __why__ understanding these differences are important for our understanding of marine $N_2$ fixation, especially within a predicted hotspot for $N_2$ fixation.

Pg 5 lines 24-25. Could you briefly outline why performing 'at sea' quantitation is/would be a preferential application for qPCR studies?

Materials and Methods:

Pg 6 21-24. Could you indicate here (briefly) which conditions you were aiming for with these LD stations?

Pg 7 line 3. Which make and model of CTD was used?

Pg 7 lines 9-11. Where these samples also stored with the glass bead mixture? Was the same amount of seawater filtered?

Pg 7 line 16. How long were they stored for?

Pg 7 line 22. Why are these methods included under the 'Nutrient analyses' sub-heading?

Pg 8 line 14. This is T. pelagicum in Supplementary Table 5?

Pg 8 line 16. Were the host diatoms quantified too? If not, why not?

Pg 9 lines 2-4. Have you tested the two DNA extraction methods on identical samples to determine any potential differences between the two methods? It would be good to get a clear sense of how different these extraction methods are here.

Pg 9 line 14. What is the percent identity between the UCYN-A1 and UCYN-A2 host 18S rRNA sequences?

Pg 9 lines 13-21. This information might be better summarised in a table.

Pg 10 lines 10-12. It's great to see this information here but why wasn't het-3 included in the het cross reactivity tests?

Pg 11 lines 8-10. Is there a particular reason why assays weren't performed for these stations?

Pg 11 lines 22-24. What were the efficiencies of the other assays?

Results:

I think the results of the cross-reactivity tests should be moved to become section 3.2 as this is important for the interpretation of the qPCR assays.

Pg 13 line 7. Table 1 contains values for DIN- should they be bq?

Pg 13 lines 18-19. A comparison of the two DNA extraction methods is required to determine if they could have affected the qPCR results.

Pg 14 lines 1-5. But some of these differences are quite large, for example from Table S2 UCYN-A1 at LDB (10% irradiance) nifH copies at sea were $1.08 \times 10^3$ compared to bd in the lab quantified samples, and UCYN-B at SD1 was bd at sea and $> 1 \times 10^5$ for the lab based assays. This is a potentially major issue, with no clear pattern for as to why. Can you explain these results? This needs to be discussed further on pages 20-21.

Pg 14 lines 17-20. Here, and in other places throughout the results where you report depths of maximum abundances, please include the *nifH* copy numbers in the text.

Pg 15 lines 4-6. Please revise this sentence for clarity.

Pg 15 lines 7-11. The different LD and SD stations within the MA and SG become a bit confusing throughout the results. Perhaps indicate the different regions in Figure 1 and supplementary tables where applicable.

Pg 15 line 14. Sometimes you refer to number of stations and other times the number of samples when talking about prevalence of the different groups, please be consistent.

Pg 16. Section 3.4 indicate number of observations included when presenting the significant correlations.

Pg 17 line 18. Perhaps indicate the significant clustering for group 1 and 2 on Figure 3 for clarity.

Pg 18 lines 1-11. The RDA is explained very nicely, perhaps you should colour code the dots in figure 4a to reflect the different response variables.

Pg 18 Section 3.5. I think the results of the meta-analysis would be more compelling if represented as a figure in the main text, perhaps as a heatmap/correlogram like Figure 3.

Pg 19. Lines 14-24. I suggest moving this section to 3.2 of the results. Do you have the data for the "vice versa" e.g. for the UCYN-A2 host assay with UCYN-A1 host target, and the het-2 assay with het-1 target? This is not obvious from Supplementary Figure 1.

Discussion:

Pg 21 lines 2-8. Please discuss these results more thoroughly. Specifically, can you comment on potential differences in DNA extraction efficiency between the two methods? Are you comparing the same diazotroph community (e.g. from the same Niskin bottle/ homogenised samples)? Were there any inhibitors? There doesn't appear to be a clear pattern in over/under estimated of the at sea versus lab assays based on Table S2, so perhaps you can't explain the differences, but possibilities should at least be discussed.

Pg 21 lines 23-25. Do you have a hypothesis as to why you observed these surprising results?

Pg 22. Could you also compare the actual abundances throughout these paragraphs to give more context- perhaps also the seasonal timing of the different studies for comparison.

Pg 24 lines 1-8. Indicating that DDAs are important for export production in this region, like the NPSG.

Pg 24 lines 9-18 Could this also be due to a limited understanding/representation of UCYN-C diversity; how specific is the qPCR assay?

Pg 28 lines 14-22. Why do you think this was the case? What other factors (perhaps beyond what you measured) could have influenced the depth distributions of these groups.

Pg 28-29 lines 23-19. It would be nice to see further discussion around the results of the meta-analysis; the similarities and differences to other regions and the local/environmental factors driving these patterns could be discussed.

Pg 30 line 19. Could you provide the same context for the UCYN-A assays?

Figures

Figure 1. Does the white dashed line indicate the separation between the MA and SG? Please clarify. Would it be possible to overlay SST on the station map, as this was an important explanatory variable.

Figure 2. You mention specific depths in the text- perhaps indicate average depth on 2a, or include % surface irradiance in the text. Can you make 2b slightly larger as the station numbers are difficult to distinguish (perhaps also indicate the MA to SG transition).

Figure 3. Indicate group 1 and group 2 on the hierarchical clustering for clarity.

Figure 4. Colour coordinated dots might help to support the text. Please include a y-axis label in 4b.

I would also like to see the meta-analysis presented as a figure if possible.

A T-S plot as a supplementary figure would also help to distinguish the different water masses of the MA and SG.

Technical corrections

Pg 3 line 21. Insert comma after 'genera'

Pg 3 line 25. Remove additional 'the'

Pg 5 line 20. Typo 'recognized'

Pg 7 line 23. 'in' rather than 'and'

Pg 7 line 24. Remove 'two different days' and parentheses.

Pg 8 line 8. Remove 'in'

Pg 8 line 16. Unclear if there is a comma missing or parentheses missing, please check.

Pg 8 line 7. Semi-colon missing

Pg 13 line 1. OpenMEE reference missing

Pg 14 line 6. 'Horizontal and vertical distributions' should be a new section

Pg 14 line 16. Remove extra 'for'

---

## Author Comment (AC1) · 6 Jul 2017

The authors would like to thank referee 1 for helpful and insightful comments for the improvement of our manuscript. Attached as a supplementary file (zip) are the extensive answers to the referee 1 comments as well as a revised version of the manuscript (including track changes for easy overview). Additionally, together with this author comment are all revised figures provided.

Please also note the supplement to this comment:
https://www.biogeosciences-discuss.net/bg-2017-63/bg-2017-63-AC1-supplement.zip

[Figure]

[Figure]

[Figure]

[Figure]

**Fig. 1.**

[Figure]

**Fig. 2.**

[Figure]

**Fig. 3.**

[Figure]

**Fig. 4.**

[Figure]

**Fig. 5.**

---

## Referee Comment (RC2) · Anonymous Referee #2 · 14 Nov 2017

**General comments**

This manuscript discusses the distribution and environmental drivers of cyanobacterial diazotrophs on a transect in the Western Tropical South Pacific. The nifH genes of major cyanobacterial diazotroph groups (and the UCYN-A prymnesiophyte hosts) were quantified via qPCR and correlated to environmental parameters. Additionally, a meta-analysis was performed to test whether the environmental drivers identified in this study were found in other ocean regions.

The authors provide useful data on the distributions of cyanobacterial diazotrophs in a historically understudied ocean region, along with the environmental context of the stations sampled. I found the environmental correlations and cluster analysis particularly compelling, and was interested in the finding that the two UCYN-A sublineages occupied a deeper zone in the water column than the other diazotroph groups investigated. However, I have a few serious concerns about this manuscript. First, the authors suggest that their finding of lower abundances of UCYN-A host 18S rRNA genes than cyanobacterial UCYN-A nifH genes may imply that the UCYN-A in their samples were in a free-living state. This is completely speculative without microscopic evidence, and these statements should be removed from the abstract and conclusion. Second, I am concerned about the large discrepancies in qPCR abundance date between separate lab-based and ship-based methods. I elaborate on all of these concerns in the specific comments below. The manuscript also contained numerous grammar errors; I correct some but not all of these errors in the technical comments below. The language in this manuscript should be improved before publication.

The page and line numbers below refer to the revised manuscript that the authors submitted after incorporating responses from the first reviewer.

**Specific comments**

P2L11-L12: Your assertion that the detection of UCYN-A nifH genes but not the host 18S rRNA genes (via your specific qPCR primer sets) may imply a free-living state for UCYN-A is highly speculative and inappropriate for the abstract. Remove this statement (I suggest removing this entire sentence).

P2L18: "temperature seemed to have a major impact": please clarify/rephrase.

P5L21: Is 17 cells per mL really a high concentration? Perhaps replace "high" with "moderate."

P6L13: Rephrase "underlying factors." Environmental drivers?

P6L17: Didn't you also target UCYN-C?

P13L1: Did you use data from both the lab-based and ship-based qPCR assays for your correlations? I find this concerning since you saw such large differences between lab and field assays.

P15L4: "we considered only when there was at least one order of magnitude difference in detection" —please clarify. I counted 38 rows in your Supp. Table 2. Does this mean that 38 out of the 44 samples for which you can make the lab-based/sea-based qPCR comparisons had over an order of magnitude difference in nifH copy numbers? I find this very concerning if you are combining the 2 datasets for your statistical tests.

Discussion overall: The discussion could be greatly streamlined, particularly section 4.3.

P22L9-P23L8: I am concerned about the large differences you observed between the qPCR performed in the lab and at sea. The supplement to this manuscript only included Supp. Fig 1 and Supp. Tables S1-S6, so I cannot see the Supplementary Figure 3 referred to in the text, which apparently addresses the inconsistencies. You say that you cannot discount the "natural heterogeneity of plankton," but it seems you could easily distinguish between natural heterogeneity and differences due to extraction/qPCR method by looking at the variability in nifH copy number among biological replicates processed in the same way. Did you include any biological replicates, taken from the same niskin, and process the samples using the same methods? If you saw the same variability among replicates as you do between the two different methods, then you could attribute the differences you see to natural variability. But if the difference in methods is the reason you see such large differences in samples processed in lab vs at sea, then perhaps you should only use one or the other dataset instead of combining them.

P23L23: Here and elsewhere, clarify that these were the least detected diazotrophs of those targeted (since you did not asses total diazotroph diversity).

P25L20: "12m, which is shallower than the subsurface maximum"—Did these studies really all compare 12m to 25m? If not, this statement should be removed.

P27L4-7: And because the UCYN-A genome suggests that it does not have the genetic capacity for independent carbon metabolism.

P27L7-17: You have already described reasons why we do not think that "the UCYN-A lineages can live freely." As you explain, the most likely reason that you found higher abundances of UCYN-A1 nifH genes than the host 18S rRNA genes is that the qPCR primers used to not cover the full diversity of the hosts. Also, you don't know that the hosts were "absent" from your samples, they were just below detection. I think it is inappropriate to speculate that UCYN-A may be free-living when you are only presenting qPCR data. You should present microscopic evidence (CARD-FISH) if you are going to make a claim that UCYN-A can exist in a free-living state.

P27L22: "we found evidence that there are multiple UCYN-A1 and A2 symbionts in both host types"— again, you need microscopic evidence to make these types of statements. The fact that you found higher abundances of UCYN-A nifH genes than the host 18S rRNA genes likely reflects that the qPCR primer/probe set does not hit the full diversity of hosts. The discussion on numbers of UCYN-A per host is entirely speculative when you only have qPCR data, so this entire paragraph should be removed or greatly shortened.

P28L1: Here and elsewhere: UCYNA-1 and A2 nifH genes were 2-10… inefficient DNA extractions, polyploidy, etc mean that nifH gene copies do not correspond to cell concentrations (as you discuss later).

P29L4-5: Also see Luo et al. (2014), Biogeosciences. I find it curious that you do not discuss this paper.

P30L21: "it would appear that low light was a pre-requisite"— this is an over-statement. You just found a correlation.

P31L13-14: Comment on the negative correlation of UCYN-A with depth in the meta analysis?

P32L9: You don't have to assume one gene copy per cell when you discuss qPCR data, as long as you refer to gene copies instead of cell abundances (e.g. UCYN-A1 nifH gene abundances instead of UCYN-A1 abundances). But throughout the manuscript, you talk discuss the concentrations of diazotrophic groups, not their gene copies. I think you should either make changes throughout the manuscript to refer to gene copies instead of cells, or else here (page 32) be explicit that YOU are assuming one gene copy per cell in this manuscript, though you realize that this assumption is likely not valid because of problems including polyploidy and inefficient extraction efficiency.

P34L11: "reliable quantification"— really?

Fig. 2:
- Did the light really attenuate the same at all of the stations?
- Clarify in the legend whether 2b depicts surface concentrations. If so, can you add error bars from biological replicates
- Capitalize depth, station etc
-  Rotate the text in 2b
- 1b is missing its panel label
- I think this figure would be easier to digest if you switched the axes in 2b and lined up the two panels vertically

Fig. 4

- It is not apparent to me what the individual points on this plot represent. Perhaps you could elaborate on the meaning of "unconstrained response variables." Or else just realize that not everyone will follow.
- Rephrase "variance of included parameters" in the figure legend

Fig. 5
Please clarify whether this analysis used all of the data included in Supp. Table 6.

**Technical comments**

P2L6-8: *"Trichodesmium*…respectively": Rephrase this sentence to improve grammar.

P2L14: Replace "deep dwelling" with "a deep-dwelling group"; replace "surface group" with "a surface group"

P3L15: Replace the comma after "surface" with a semicolon.

P3L19: Replace "photic" with "photic-zone"

P4L2: Replace "is a symbiosis between" with "associates with"

P4L12: Replace "the UCYN-C" with "the UCYN-C group"

P5L9: Replace "lowest concentrations" with "lowest reported concentrations" and delete "in the world have been reported"

P5L10-11: replace "harboring" with "which harbors" and replace "being" with "is"

P6L2: Place a comma after "WTSP" and replace the semicolon with a comma.

P7L6-9: This seems to repeat the sentence P6L24-P7L3.

P9L6: Returned to the laboratory AND frozen? Please clarify.

P9L22: Replace "on published 18S rRNA sequence" with "on a published" or "on published…sequences"

P10L18: Replace "selected diazotrophs nifH gene copies" with "nifH gene copies from selected diazotrophic groups"

P10L20: Replace "performed" with "quantified"

P11L17: Include the end parentheses after "Biosystems"

P13L10-13: "T-tests…concentrations" I find this sentence confusing.

P13L13: Replace "dataset" with "data"

P14L8 "but declined…compared to the SG" Be more specific.

P14L9-13: Rephrase this sentence.

P19L3-5: Rephrase this sentence.

P19L12: Replace "The deeper dwelling" with "Diazotrophic targets in the deeper dwelling"

P20L25: Replace "and significantly" with "but was significantly"

P21L20-21: "and likely… N2 fixation" please rephrase.

P22L2: Rephrase "nowadays"

P22L6: Replace "showed" with "describe." Also, I think the term efficient is inappropriate, as you did not measure DNA extraction efficiency.

P22L7: Replace "qPCR" with "qPCR technique"

P24L3: Replace "to Moisander's" with "to that reported by Moisander"

P24L8: "symbioses"—you mean both A1 and A2? Please clarify.

P24L25: Replace "lesser" with "lower"

P25L19: Replace "Highest" with "The highest"

P28L11: Replace "ranging" to "ranging from"

P28L14: Replace "conditions" with "the conditions"

P28L15: Replace "life histories" with "life histories of different diazotrophic groups"

P29L1-3: Rephrase this sentence to fix grammar errors.

P29L9: Here and elsewhere: replace "diazotrophs" with "diazotrophic groups"

P29L12: Replace "environmental parameters PAR" with "the environmental parameters of PAR"

P29L13: "influencing", "drove"—here and elsewhere, rephrase so you are not inferring causation.

P29L19: Replace "are" with "have been"

P30L1-8: These sentences don't fit with the rest of the paragraph.

P30L12-14: "Moreover…Karl et al. 2012"— This sentence doesn't fit with the rest of the paragraph.

P30L19: Replace "and a negative" with "and displayed a negative"

P30L24: "Interesting and unexpected was"— rephrase.

P31L5: Replace "diazotrophs" with "diazotroph"

P31L11-13: "The studies…temperature" correct the grammar errors in this sentence.

P32L3-6: "Unlike…space" perhaps delete this sentence.

P33L9: "Consistent…abundant"— rephrase, this statement is meaningless out of context.

P33L17-21: "According…tests"— remove this sentence.

---

## Author Comment (AC2) · 22 Nov 2017

The authors would like to thank referee 2 for insightful and helpful comments and suggestions to improve our submitted manuscript. Attached as a zipped supplementary to this author comment are the extensive answers to referee 2 comments (pdf), the revised manuscript (pdf; including track changes) and the suppl. figures and tables (excel-file). Further down this author comment, the revised fig02 is provided.

Please also note the supplement to this comment:
https://www.biogeosciences-discuss.net/bg-2017-63/bg-2017-63-AC2-supplement.zip

[Figure]

**Fig. 1.**

---

## Author Response (AR1)

**Answers to comments from reviewer 1**

Here we respond to the reviewer comments/suggestions beneath and in italicize.
We have updated the appropriate line and page numbers.

Note: Since we only had a pdf of the reviewer suggestions we address the 3
paragraphs from the general comments without typing all text from the reviewer then
address the specific comments below.

**General Comments.**

**Surprise about the picoeukaryote.**

*This is a valid suggestion however at the time of the study the sequence data for the
respective diatom hosts was unknown. A subsequent and current study in Foster lab is
work developing new molecular methods (e.g. in situ hybridization, qPCR) for
identifying on the diatom hosts by single cell PCRs, however this information was not
concurrent with the work presented here so it is not included. Unfortunately we also
have no more DNA template left to process as we used it all up on the 9 targets which
we report on.*

**Discrepancies between 'at sea' and lab-based.**

*The comments and suggestions (see below) are also well noted, however we cannot
address the extraction efficiencies given the nature of the samples (field based mixed
communities. Although we attempted to be as uniform as possible in the sampling and
archiving, it is likely that some heterogeneity in sampling resulted from that samples
were take from different levels of the niskin. Perhaps all water could have been
drained, homogenized prior to the subsampling of the volumes for the nucleic acids,
however we were pressed to get the samples extracted and run on the qPCR in a
timely manner to direct other points of interest for the cruise at large. We address
some of the sources for the discrepancies in more detail in the revised version (Pgs
22-23, lines 13-4).*

**Restructuring and refining of introduction, results, etc.**

*We agree, and have made some reorganizing of the presented work.*

*It should be noted that this paper was submitted for a special issue of Biogeosciences
which is entirely focused on the research cruise (OUTPACE) and so it is part of a
collective body of work by all members of the expedition.*

**Specific Comments**

**Abstract:**

**In general, the Abstract could be improved by including quantitative nifH copy number rather than the percentage of nifH copies detected.**

*Agreed, and have provided ranges in the nifH copies enumerated for the various
targets.*

**Line 9. What does the >47% refer to when you say the UCYN-A lineages were poorly detected?**

*Originally this was referring to the % of samples below detection, we have rewritten the sentence and this no longer applies (Pg 2, Lines 10-12).*

**Line 9-12. This is inconsistent-the hosts mirrored their respective symbionts yet were below detection?**
*We recognize that this was not very clear as we were referring to when the hosts were present they were only in samples where their respective symbionts were enumerated. We have re-written the result (Pg 2, Lines 13-15).*

**Lines 14-15. Perhaps re-phrase to state that they correlated with the surface group. Include the parameters that were significantly correlated with the deep UCYN-A group too.**
*Agreed and have followed the suggestions (Pg 2, Lines 19-20).*

**Lines 15-16. Could you expand on this briefly?**
*Added in a short comment on the most important environmental parameter in the meta-analysis: temperature (Pg 2, Line 18).*

**Line 18. 'free-living cyanobacterial diazotrophs'**
*Agreed and rephrased (Line 2, Line 20)*

**Introduction:**
**Pg 4 lines 18-19. There is no clear link between these 2 paragraphs. Suggest moving.**
*Agreed and have restructured the introduction following these helpful suggestion.*

**Pg 5 lines 18-23. Perhaps expand on this slightly to indicate why understanding these differences are important for our understanding of marine N₂ fixation, especially within a predicted hotspot of N₂ fixation.**
*Added a sentence to emphasize the importance and benefits of a better understanding (Pg 5, Lines 3-5).*

**Pg 5 lines 24-25. Could you briefly outline why performing 'at sea' quantitation is/would be a preferential application for qPCR studies.**
*We have added a brief explanation as to why it was necessary and useful for the expedition. For example, the results from the qPCR was used to make informed decisions about regions high for target diazotrophs. Subsequently these stations were selected for long duration stations so intensive sampling could be implemented to better characterize the community, the microbial activities and hydrographic conditions (e.g. rate measures, drifter arrays, sediment traps, etc.) (Pg 6, Lines 3-8).*

**Materials and Methods:**

**Pg 6 21-24. Could you indicate here briefly which conditions you were aiming for with these LD stations.**
*A brief explanation is provided and also two articles in the special issue are referenced (Pgs 6-7, Lines 25, 1-4).*

**Pg 7 line 3. Which make and model of CTD was used.**
*The Seabird 911 was added (Pg 7 Line 15).*

**Pg 7, lines 9-11. Were these samples also stored with the glass bead mixture. Was the same amount of seawater filtered?**

*Details have been provided. The 4 samples taken for the 'at sea' processing were not stored, these were immediately extracted and the beads were necessary for the agitation step. The same volume was always filtered (2.5L), except for a few cases with a lot of material clogging the filters (Pg 7 Lines 19-20; Pg 8, Lines 1-2).*

**Pg 7 line 16. How long were they stored?**

*The nutrient analysis samples were stored for a few months as all samples were re-analyzed in the land-based laboratory (Pg 9, Lines 7-8).*

**Pg 7 Line 22. Why are these methods included under the 'Nutrient analyses' sub-heading?**

*This was a mistake, the text has been replaced in an appropriate subheading (Pg 7, Line 11).*

**Pg 8 line 14. The *T. pelagicum* in Suppl. Table 5?**

*This was also a mistake, both should be* K.pelagicum *and have now been corrected.*

**Pg 8. Line 16. Were the host diatoms quantified too? If not why not?**

*The host diatoms were not quantified as their 18S rRNA sequences were unknown at the time of the study and hence qPCR primers and probes could not be designed. Subsequent and current work in Foster's Lab is working on these types of assays. Unfortunately all DNA templates were used up on the 9 targets (Trichodesmium, UCYNA-1, UCYNA-2, UCYNA-1 host, UCYNA-2 host, UCYN-B, het-1, het-2, het-3).*

**Pg 9 Lines 2-4. Have you tested the two DNA extraction methods on identical samples to determine any potential differences between the two methods? It would be good to get a clear sense of how different these methods are here.**

*We wholly agree with this shortcoming in our approach, but unfortunately we do not have replicate samples to test. It should also be noted that replicate field samples are challenging to collect as one cannot be certain of uniform distribution in a niskin bottle. We have added a short summary of the latter in the discussion (Pg 22-23 Lines 13-4).*

**Pg 9 Line 14. What is the percent identity between the UCYN-A1 and UCYN-A2 host 18S rRNA sequences?**

*We have added this detail (97.95%) by determining a distance matrix on the 635bp fragment of the 18S rRNA for the following sequences: accession number JX291893 (UCYN-A1 Host) and accession numbers KF771248-KF771254 (UCYN-A2 host) (Pg 10 Lines 16-17).*

**Pg 9 Lines 13-21. This information might be better summarized in a table.**

*Agreed and have added an additional suppl. table 1 (1c).*

**Pg 10 lines 10-12. It's great to see this information here but why wasn't het-3 included in the het cross reactivity tests?**

*Earlier cross-reactivity tests reported in Foster et al. 2007 found no cross-reaction between the het-3 and the other het groups, so we felt it was redundant to repeat. This detail is added into the text. (Pg 11, Lines 6-8).*

**Pg 11 lines 8-10. Is there a particular reason why assays weren't performed for these stations?**
*Overall qPCR assays 'at sea' were limited by time. The main reason for not performing qPCR at these stations was that they were not possible LD station candidates due to geographical and/or hydrological reasons. A short sentence has been added in the text (Pg 12 Line 5).*

**Pg 11 lines 22-24. What were the efficiencies of the other assays?**
*The efficiencies were only tested on the het groups as we were trying to be conservative with the limited template (Pg 12 Line 19)*

**Results:**

**I think the results of the cross-reactivity tests should be moved to become section 3.2 as this is important for the interpretation of the qPCR assays.**
*Agreed and have moved the text. Although it should be noted that the intention of this paper was not to be a 'methods' paper, and we are subsequently working on developing better qPCR assays which will be summarized in a separate body of work at a later date.*

**Pg 13 Line 7. Table 1 contains values for DIN-should they be bq?**
*We acknowledge the inconsistency and have amended the table where the bq is defined.*

**Pg 13 Lines 18-19. A comparison of the two DNA extraction methods is required to determine if they could have affected the qPCR results.**
*We agree with the statement that a comparison of the extraction method is necessary if one was to truly compare the results. However here our intention is to report the differences, which we attained. The differences in abundance could be derived from the difference in extraction method, the variation of extraction efficiency per target, the patchiness of plankton, etc. It is not clear how the reviewer suggests we correct this section.*

**Pg 14 Lines 1-5. Explanation for the differences in the at sea and the lab based. Can you explain these results? This needs to be discussed further on pages 20-21.**
*There are a few possibilities here, and are explained in the text (pg 22-23, Lines 20-25; Lines 1-11).*

**Pg 14 Lines 17-20. Here and in other places throughout the results where you report depths of maximum abundances, please include the *nifH* copy numbers in the text.**
*Agreed and have modified the text throughout section 3.4. Note that the depth of maximum abundances is the average from the two regions (MA and SG).*

**Pg 15. Lines 4-6. Please revise this sentence for clarity.**
*Agreed and have made it several sentences. (Pg 16, Lines 16-19).*

**Pg 15 Lines 7-11. The confusion with the LD and SD stations within the MA and SG.**

*Agreed and have followed the suggestion of the reviewer to indicate on Figures and also in the suppl. table 3.*

**Pg. 15 line 14.Sometimes you refer to number of stations and other times the number of samples when talking about prevalence of the different groups, please be consistent.**

*Noted and rephrased where appropriate. Although, we include the detail on samples when referring to the UCYN-A as it was striking that these were absent and/or patchy in distribution and we lose this context if we only refer to station.*

**Pg. 16 Section 3.4 indicate number of observations included when presenting the significant correlations.**

*Noted and fixed in the text.*

**Pg. 17 line 18.  Perhaps indicate the significant clustering for group 1 and 2 on Figure 3 for clarity.**

*Agreed and amended the Figure with this detail.*

**Pg 18 lines 1-11. The RDA is explained very nicely, perhaps you should color code the dots in figure 4a to reflect the different response variables.**

*Figure 4a, the dots in the RDA represent samples and cannot be color coded according to diazotrophs, since they are represented by a vector (red labeled dot).*

**Pg 18 Section 3.5. I think the results of the meta-analysis would be more compelling if represented as a figure in the main text, perhaps as a heatmap/correlogram like Figure 3.**

*Agreed, we have added bar graphs (one for each diazotroph) as an additional figure (Figure 5).*

**Pg 19 Lines 14-24.  I suggest moving this section to 3.2 of the results. Do you have the data for the "viceversa" e.g. UCYN-A2 host assay with the UCYN-A1 host target, and the het-2 assay with the het-1 target? This is not obvious from Supplementary Figure 1.**

*We did run the similar assays in the reverse ('vice versa') and have added the additional graphs in supplementary figure 1. Originally we limited the presentation since the results were similar and we were trying to keep things simplified. (Pg 14, section 3.2)*

**Discussion:**

**Pg 21 lines 2-8. Please discuss these results more thoroughly.**

*Agreed and have added text (Pg 22-23, Lines 13-4).*

**Specifically, can you comment on potential differences in DNA extraction efficiency between the two methods?**

*Agreed that this would be a valuable piece of information however we did not determine the DNA extraction efficiency, and it would also be quite difficult since*

*these are mixed community field samples. One would need a known abundance of a particular target e.g. lab based culture work could address this for a particular target that has been isolated e.g. UCYN-B and* Trichodesmium.

**Are you comparing the same diazotroph community (e.g. from the same Niskin bottle/homogenized samples).**
*Samples were always taken from the same niskin bottle, but their entire volume was not homogenized.*

**Were there any inhibitors?**
*No inhibitors were added.*

**There doesn't appear to be a clear pattern in over/under estimated of the at sea versus lab assays based on Table S2, so perhaps you can't explain the differences, but possibilities should be at least discussed.**
*Agreed and further details have been added to the text (Pg 22 Lines 19-23).*

**Pg 21. Lines 23-25. Do you have a hypothesis as to why you observed these surprising results?**
*Assuming that the reviewer is referring to the restriction of the UCYN-A to one depth of LD C, perhaps one could hypothesize that it was a small-scale bloom or entrainment along an isocline. However we prefer to be conservative since it was only one observation and leave the text as is without drawing a larger conclusion.*

**Pg 22. Could you also compare the actual abundances throughout these paragraphs to give more context-perhaps also the seasonal timing of the different studies for comparisons?**
*We are not sure how to address this suggestion. Is it that we should provide the exact nifH copy numbers for all the 11 datasets?*

**Pg 24 lines 1-8. Indicating that DDAs are important for export production in this region, like the NPSG.**
*Agreed, added a sentence (Pg 26 Lines 13-14).*

**Pg 24. Lines 9-18. Could this also be due to a limited understanding/representation of UCYN-C diversity; how specific is the qPCR assay.**
*The cited studies for UCYN-C have used qPCR (Turk-Kubo et al., 2015) and isolation, then subsequent 16S rRNA and nifH sequencing (Taniuchi et al., 2012). The qPCR study, which was conducted within a lagoon of the Melanesian archipelago, found UCYN-C to partly dominate diazotroph abundances, and they used the same qPCR assay as we did (Foster et al., 2007). We have added a short summary statement about the UCYN-C assay, which was evaluated in Turk-Kubo et al. 2015 that reported the UCYN-C assay to quantify the majority (up to 85% of the sequences in their study) of UCYN-C phylotypes (Pg 26, Lines 23-24).*

**Pg 28 lines 14-22. Why do you think this was the case? What other factors (perhaps beyond what you measured) could have influenced the depth distributions of these groups.**

*UCYN-A has been shown to have a colder temperature optimum (and range) than the other cyanobacterial diazotrophs in this study, which could also drive a subsurface maximum. Moreover the distributions observed could be linked to viral infections and grazing by zooplankton, none of which was actually measured.*

**Pg 28-29 lines 23-19. It would be nice to see further discussion around these results of the meta-analysis: the similarities and differences to other regions and the local/environmental factors driving these patterns could be discussed.**
*Agreed. Added text (Pg 30-32 Lines 23-3).*

**Pg 30 Line 19. Could you provide the same context for the UCYN-A assay.**
*The cross reactivity for the two UCYN-A assays had a near perfect match when run as A2 assay with A1 standard, meaning that no matter the abundance of A1 (high or low), there would always be a risk of significant cross-reactivity and overestimation of A2 in the presence of A1.*

**Figures:**

**Figure 1. Does the white dashed line indicate the separation of the MA and SG? Please clarify. Would it be possible to overlay SST on the station map, as this was an important explanatory variable?**
*Yes, the dotted line separates the MA an SG and has been clarified in the caption. It is possible to overlay the map with SST but it will not be very informative in our relatively narrow region where the SST was mostly uniform (everything would be the same color). The large impact of temperature was mediated vertically rather than horizontally.*

**Figure 2. You mention specific depths in the text-perhaps indicate average depth on 2a or include % surface irradiance in the text. Can you make 2b slightly larger as the station numbers are difficult to distinguish (Perhaps also indicate the MA to SG transition).**
*Agreed*

**Figure 3. Indicate group 1 and group 2 on the hierarchical clustering for clarity.**
*Agreed. The two hierarchical clusters of group 1 (surface) and group 2 (subsurface) has been marked.*

**Figure 4. Color coordinated dots might help to support the text. Please include a y-axis label in 4b.**
*Agreed, however see previous reply in the results section regarding coloring the dots. Y-axis label (Variance) added.*

**I would also like to see the meta-analysis presented as a figure if possible.**
*Agreed and have added a new figure 5 which summarizes the meta-analysis.*

**A T-S plot as a supplementary figure would also help to distinguish the different water masses of the MA and SG.**
*This is a valid suggestion included a supplementary Figure 2 which has the upper 500m T-S plots for stations LD A and SD 15 (MA and SG, respectively). More details*

*on the water masses will likely be addressed by the physical oceanographers whom have submitted work on the hydrography in the special issue.*

[revised manuscript text omitted]

RDA1 and RDA2 axes meaning that most of the variance observed is explained by the included environmental parameters. The arrows are the constrained explanatory vectors with the dots representing the superimposed unconstrained response variables. PAR and nutrients (DIP and DIN) were omitted due to limited data.

**Figure 5 a-d.** Meta-analysis bar graphs, a) *Trichodesmium*, b) UCYN-B, c) UCYN-A, d) het-

1, with the significant ($p < 0.05$) parameters for each diazotroph arranged as the strongest effect to the left and weakest to the right (either positive or negative). Each parameter is color coded, where the cyanobacterial diazotrophs have been assigned a spectrum of orange from

*Trichodesmium* (darkest) to het-2 (lightest). Red=temperature, blue=salinity, black=depth, green=chlorophyll *a*, yellow=DIN and purple=DIP.

**Answers to comments from reviewer 2**

Here we respond to the reviewer comments/suggestions beneath and in italicize. We have updated the appropriate line and page numbers.

**Specific comments**

**P2L11-L12: Your assertion that the detection of UCYN-A nifH genes but not the host 18S rRNA genes (via your specific qPCR primer sets) may imply a free-living state for UCYN-A is highly speculative and inappropriate for the abstract. Remove this statement (I suggest removing this entire sentence).**

*We have deleted the speculation, and state the percent of samples where UCYNA1 and A2 were detected and their respective hosts below detection (Pg. 2, lines 11-12)*

**P2L18: "temperature seemed to have a major impact": please clarify/rephrase.**

*We have clarified our statement. (Pg. 2, Line 17-20)*

**P5L21: Is 17 cells per mL really a high concentration? Perhaps replace "high" with "moderate."**

*We have followed the reviewer suggestion and modified our text. (Pg. 5, line 21)*

**P6L13: Rephrase "underlying factors." Environmental drivers?**

*Changed according to reviewer's suggestion. (Pg. 6, line 13).*

**P6L17: Didn't you also target UCYN-C?**

*We have added the UCYN-C back in the text here, but it should be noted that the UCYN-C was only quantified in the 'at-sea qPCR' hence the comparison is difficult since we have far fewer abundance estimates, and therefore it wasn't included in any statistical analyses. (Pg. 6, line 17)*

**P13L1: Did you use data from both the lab-based and ship-based qPCR assays for your correlations? I find this concerning since you saw such large differences between lab and field assays.**

*The data generated by the 'at sea' qPCR assays was not used for any subsequent statistical analyses since only a few targets were quantified (UCYN-A1, UCYN-A2, UCYN-B, and UCYN-C) and only 4 depths at a limited number of stations. We included the 'at sea' qPCR for a comparison with the lab-based (archived) sample processing.*

**P15L4: "we considered only when there was at least one order of magnitude difference in detection" —please clarify. I counted 38 rows in your Supp. Table 2. Does this mean that 38 out of the 44 samples for which you can make the lab-based/sea-based qPCR comparisons had over an order of magnitude difference in nifH copy numbers? I find this very concerning if you are combining the 2 datasets for your statistical tests.**

*We noted in the Suppl. Table 2 only when a sample was at least 1 order higher/lower for a particular target (UCYN-A1, UCYN-A2 and UCYN-B), since these were the only targets processed in both the 'at sea' and lab-based qPCR). So for example, we detect 18 nifH copies L$^{-1}$ for UCYN-A1 in the 'at sea' sample from SD3 35 m, and in the parallel separate sample filtered at sea and stored until processing (full extraction and qPCR) in the lab was dnq. In addition, there is a suppl. Figure illustrates the comparison; this is Suppl. Fig. 3 (see attached excel file) and one can more easily see that for UCYN-B and UCYN-A1, often values fall on a 1:1 line.*

*As stated above the 2 datasets were not pooled, since the extraction protocol was not identical it was not appropriate to pool the values.*

*We have clarified this paragraph and included the correct reference to the Suppl. Figure. (Pg. 15, lines 4-9).*

**Discussion overall:** **The discussion could be greatly streamlined, particularly section 4.3.**
*We have followed the suggestion of the reviewer and have tried to shorten the discussion. Many of our results concur with previous findings, and so we try to only highlight the new contributions of this work.*

**P22L9-P23L8: I am concerned about the large differences you observed between the qPCR performed in the lab and at sea. The supplement to this manuscript only included Supp. Fig 1 and Supp. Tables S1-S6, so I cannot see the Supplementary Figure 3 referred to in the text, which apparently addresses the inconsistencies. You say that you cannot discount the "natural heterogeneity of plankton," but it seems you could easily distinguish between natural heterogeneity and differences due to extraction/qPCR method by looking at the variability in nifH copy number among biological replicates processed in the same way. Did you include any biological replicates, taken from the same niskin, and process the samples using the same methods? If you saw the same variability among replicates as you do between the two different methods, then you could attribute the differences you see to natural variability. But if the difference in methods is the reason you see such large differences in samples processed in lab vs at sea, then perhaps you should only use one or the other dataset instead of combining them.**
*Unfortunately biological replicates were not taken during this cruise, as we were pressed to process the 'at sea' qPCR in a timely manner to inform the cruise at large. Moreover, replication in sampling (e.g. 2-3 replicate samples) for qPCR is not typical (see previous works from multiple groups) and is often related to time and/or water budgets. Although, here it seems that it should be a consideration in future samplings.*

*As stated above, the datasets were not pooled, and the comparison of the qPCR from the 'at sea' and archived samples was shown to be as transparent as possible. More efforts are underway to address many of the technical details (e.g. at sea vs. archive processing, the cross-reactivity) unveiled in this body of work, however for the purposes of this manuscript and limitations in that we do not have biological replicates, we care to focus on the results in the context of the special issue-which is to provide the abundance for the diazotrophs and the influence on the measured parameters. We do acknowledge the issues here and that is why we wanted to highlight some of the discrepancies; however we would like to stress that the two datasets were not combined for statistical analysis. A sentence has been added to clarify this. (Pg 13, line 1-3)*

**P23L23: Here and elsewhere, clarify that these were the least detected diazotrophs of those targeted (since you did not asses total diazotroph diversity).**
*We agree that this is the case and have clarified according to reviewer suggestion. (Pg. 24, line 6).*

**P25L20: "12m, which is shallower than the subsurface maximum"— Did these studies really all compare 12m to 25m? If not, this statement should be removed.**

*Here we are trying to highlight that the depth of maximum abundance for the Richelia symbiont groups (het-1 and het-2) in the MA region of the transect was shallower than in the other works. In the earlier works the upper photic zone was also sampled (e.g. WTNA the 100%-0.1% light level) and the depth of maximum abundance is reported to illustrate a niche partitioning (e.g. Moisander et al. 2010 for the UCYN groups and Trichodesmium). We have replaced 'commonly' with 'previously'. (Pg. 26, line 3).*

**P27L4-7: And because the UCYN-A genome suggests that it does not have the genetic capacity for independent carbon metabolism.**
*We have modified the text to address the dependency of the UCYN-A on other organisms based on its genome content and appropriate references. (Pg. 27, lines 11-13).*

**P27L7-17: You have already described reasons why we do not think that "the UCYN-A lineages can live freely." As you explain, the most likely reason that you found higher abundances of UCYN-A1 nifH genes than the host 18S rRNA genes is that the qPCR primers used to not cover the full diversity of the hosts. Also, you don't know that the hosts were "absent" from your samples, they were just below detection. I think it is inappropriate to speculate that UCYN-A may be free-living when you are only presenting qPCR data. You should present microscopic evidence (CARD-FISH) if you are going to make a claim that UCYN-A can exist in a free-living state.**
*We agree and have removed the speculation on the free-living nature of UCYN-A. (Pg. 28, lines 5-12).*

**P27L22: "we found evidence that there are multiple UCYN-A1 and A2 symbionts in both host types"— again, you need microscopic evidence to make these types of statements. The fact that you found higher abundances of UCYN-A nifH genes than the host 18S rRNA genes likely reflects that the qPCR primer/probe set does not hit the full diversity of hosts. The discussion on numbers of UCYN-A per host is entirely speculative when you only have qPCR data, so this entire paragraph should be removed or greatly shortened.**
*We agree that microscopic evidence is required in addition to the qPCR results and have deleted our interpretation. We highlight that the broader diversity is a possibility and state that CARD-FISH is necessary. (Pg. 28, lines 5-12)*

**P28L1: Here and elsewhere: UCYNA-1 and A2 nifH genes were 2-10… inefficient DNA extractions, polyploidy, etc mean that nifH gene copies do not correspond to cell concentrations (as you discuss later).**
*We agree, and have modified the text throughout. (Pg. 28, line 5)*

**P29L4-5: Also see Luo et al. (2014), Biogeosciences. I find it curious that you do not discuss this paper.**
*We have added this reference and also a summarizing sentence to the discussion. (Pg. 29, lines 15-17).*

**P30L21: "it would appear that low light was a pre-requisite"— this is an over-statement. You just found a correlation.**
*We agree, and have changed the wording to 'correlates with'. (Pg. 31, line 6).*

**P31L13-14: Comment on the negative correlation of UCYN-A with depth in the meta-analysis?**
*We have amended our text (Pg. 31-32, lines 24-2).*

**P32L9: You don't have to assume one gene copy per cell when you discuss qPCR data, as long as you refer to gene copies instead of cell abundances (e.g. UCYN-A1 nifH gene abundances instead of UCYN-A1 abundances). But throughout the manuscript, you talk discuss the concentrations of diazotrophic groups, not their gene copies. I think you should either make changes throughout the manuscript to refer to gene copies instead of cells, or else here (page 32) be explicit that YOU are assuming one gene copy per cell in this manuscript, though you realize that this assumption is likely not valid because of problems including polyploidy and inefficient extraction efficiency.**
*As suggested, we have made changes throughout the text, when appropriate, to reflect that we're not assuming one gene copy per cell. The, limitations, caveats of qPCR has been modified as well. (Pg. 32, lines 21-22)*

**P34L11: "reliable quantification"— really?**
*We have modified this paragraph about the use and considerations of 'at sea' qPCR. (Pg. 34-35, lines 25-1).*

**Fig. 2:**
**- Did the light really attenuate the same at all of the stations?**
*No, there were slight differences except for LD B and three stations in the SG. The caption has been modified. (Pg. 47, lines 7-13).*

**- Clarify in the legend whether 2b depicts surface concentrations. If so, can you add error bars from biological replicates?**
*The figure caption states that both a) and b) are LOG10 transformed mean concentrations across the entire cruise at a) depth and b) station. Due to heterogeneity of the diazotrophs abundances, especially with depth (where there can be 10^5 gene copies/L at the surface and 0 at 80 m), error bars would be large and not very informative in this context and so these were omitted. (Pg. 47, lines 7-13)*

**- Capitalize depth, station etc.**
**- Rotate the text in 2b**
**- 1b is missing its panel label**
**- I think this figure would be easier to digest if you switched the axes in 2b and lined up the two panels vertically.**
*We agree and have modified the figure according to reviewer suggestions. (see author comment pdf)*

**Fig. 4**
**- It is not apparent to me what the individual points on this plot represent. Perhaps you could elaborate on the meaning of "unconstrained response variables." Or else just realize that not everyone will follow.**
*We tried to amend the figure caption for clarity. (Pg. 47, lines 18-24)*

*- **Rephrase "variance of included parameters" in the figure legend.***
*We have followed the suggestion and amended the figure caption for clarity. (Pg. 47, lines 18-24).*

**Fig. 5**
**Please clarify whether this analysis used all of the data included in Supp. Table 6.**
*We have modified the figure caption and include a statement that the figure is based on Suppl. Table 6. (Pg. 48, lines 1-6).*

**Technical comments**

**P2L6-8:** ***"Trichodesmium…respectively":* Rephrase this sentence to improve grammar.**
*The sentence has been slightly changed to improve clarity and grammar (Pg. 2, lines 6-8).*

**P2L14: Replace "deep dwelling" with "a deep-dwelling group"; replace "surface group" with "a surface group"**
*Replaced according to reviewer suggestion. (Pg. 2, lines 13-14).*

**P3L15: Replace the comma after "surface" with a semicolon.**
*Replaced according to reviewer suggestion. (Pg. 3, line 15).*

**P3L19: Replace "photic" with "photic-zone"**
*Replaced according to reviewer suggestion. (Pg. 3, line 19).*

**P4L2: Replace "is a symbiosis between" with "associates with"**
*Replaced according to reviewer suggestion. (Pg. 4, line 2).*

**P4L12: Replace "the UCYN-C" with "the UCYN-C group"**
*Replaced according to reviewer suggestion. (Pg. 4, line 14).*

**P5L9: Replace "lowest concentrations" with "lowest reported concentrations" and delete "in the world have been reported"**
*Replaced and deleted according to reviewer suggestions. (Pg. 5, lines 8-9).*

**P5L10-11: replace "harboring" with "which harbors" and replace "being" with "is"**
*Replaced according to reviewer suggestions. (Pg. 5, line 10).*

**P6L2: Place a comma after "WTSP" and replace the semicolon with a comma.**
*Replaced and added according to reviewer suggestion. (Pg. 6, line 2).*

**P7L6-9: This seems to repeat the sentence P6L24-P7L3.**
*We have modified the sampling section to limit redundancy. (Pg. 6-7, lines 21-7).*

**P9L6: Returned to the laboratory AND frozen? Please clarify.**
*This sentence was unclear and we have rephrased it for clarity. (Pg. 9, line 6).*

**P9L22: Replace "on published 18S rRNA sequence" with "on a published" or "on published…sequences"**
*Replaced according to reviewer's first suggestion. (Pg. 9, line 22-23).*

**P10L18: Replace "selected diazotrophs nifH gene copies" with "nifH gene copies from selected diazotrophic groups"**
*Replaced according to reviewer suggestion. (Pg. 10, line 18).*

**P10L20: Replace "performed" with "quantified"**
*Replaced according to reviewer suggestion. (Pg. 10, line 20).*

**P11L17: Include the end parentheses after "Biosystems"**
*Added according to reviewer suggestion. (Pg. 11, line 17).*

**P13L10-13: "T-tests…concentrations" I find this sentence confusing.**
*The sentence has been amended for clarity (Pg. 13, lines 12-13).*

**P13L13: Replace "dataset" with "data"**
*Replaced according to reviewer suggestion. (Pg. 13, line 14).*

**P14L8 "but declined…compared to the SG" Be more specific.**
*We have modified the sentence to describe the deepening of the thermocline in the SG compared to the MA (Pg. 14, lines 7-9)*

**P14L9-13: Rephrase this sentence.**
*The sentence has been split and amended for clarity. (Pg. 14, lines 9-13).*

**P19L3-5: Rephrase this sentence.**
*The sentence has been amended for clarity. (Pg. 19, lines 10-12).*

**P19L12: Replace "The deeper dwelling" with "Diazotrophic targets in the deeper dwelling"**
*We have rephrased the sentence and refer to the 2 groups as shallow and deep to avoid confusion. So we refrain from using deeper and shallower, etc. (Pg. 19, lines 9-22)*

**P20L25: Replace "and significantly" with "but was significantly"**
*Replaced according to reviewer suggestion. (Pg. 21, line 6).*

**P21L20-21: "and likely… $N_2$ fixation" please rephrase.**
*Rephrased according to reviewer suggestion. (Pg. 21-22, lines 25-3).*

**P22L2: Rephrase "nowadays"**
*Wording changed to 'modern'. (Pg. 22, line 9).*

**P22L6: Replace "showed" with "describe." Also, I think the term efficient is inappropriate, as you did not measure DNA extraction efficiency.**
*We agree and have replaced the words according to reviewer suggestion. (Pg. 22, line 13).*

**P22L7: Replace "qPCR" with "qPCR technique"**
*Replaced according to reviewer suggestion. (Pg. 22, line 14).*

**P24L3: Replace "to Moisander's" with "to that reported by Moisander"**
*Replaced according to reviewer suggestion. (Pg. 24, line 10).*

**P24L8: "symbioses"—you mean both A1 and A2? Please clarify.**
*Yes, we mean both A1 and A2 and have amended this to the sentence. (Pg. 24, line 14).*

**P24L25: Replace "lesser" with "lower"**
*Replaced according to reviewer suggestion. (Pg. 25, line 6).*

**P25L19: Replace "Highest" with "The highest"**
*Replaced according to reviewer suggestion. (Pg. 26, line 1).*

**P28L11: Replace "ranging" to "ranging from"**
*Replaced according to reviewer suggestion. (Pg. 28, lines 22-23).*

**P28L14: Replace "conditions" with "the conditions"**
*Replaced according to reviewer suggestion. (Pg. 28, line 25).*

**P28L15: Replace "life histories" with "life histories of different diazotrophic groups"**
*Replaced according to reviewer suggestion. (Pg. 29, line 1).*

**P29L1-3: Rephrase this sentence to fix grammar errors.**
*This sentence has been modified and streamlined. (Pg. 29, Lines 12-13).*

**P29L9: Here and elsewhere: replace "diazotrophs" with "diazotrophic groups"**
*Since in our statistical analyses we find two groups (deep and shallow) we refrain from using diazotrophic group to not confuse.*

**P29L12: Replace "environmental parameters PAR" with "the environmental parameters of PAR"**
*We agree and have modified as suggested. (Pg. 29, lines 23-24).*

**P29L13: "influencing", "drove"—here and elsewhere, rephrase so you are not inferring causation.**
*We agree and have modified the text when appropriate (Pg. 29, line 25)*

**P29L19: Replace "are" with "have been"**
*We agree and have replaced the wording (Pg. 30, line 6).*

**P30L1-8: These sentences don't fit with the rest of the paragraph.**
*This paragraph has been modified to highlight that the nutrient conditions in this region favored DDAs over the UCYN-A. (Pg. 30, lines 3-20)*

**P30L12-14: "Moreover…Karl et al. 2012" - This sentence doesn't fit with the rest of the paragraph.**
*Our intention of including Karl et al. 2012 was to highlight that day length which is in the context of light, could influence the symbiotic diatom populations. For the sake of streamlining, the sentence has been removed.*

**P30L19: Replace "and a negative" with "and displayed a negative"**
*We agree and have modified accordingly (Pg. 31, line 4).*

**P30L24: "Interesting and unexpected was"— rephrase.**
*The sentence has been modified. (Pg. 31, lines 8-10).*

**P31L5: Replace "diazotrophs" with "diazotroph"**

*Replaced according to reviewer suggestion. (Pg. 31, line 15).*

**P31L11-13: "The studies…temperature" correct the grammar errors in this sentence.**
*We agree that this sentence was broken, and have amended and changed for clarity (Pg. 31, lines 21-23)*

**P32L3-6: "Unlike…space" perhaps delete this sentence.**
*We prefer to keep this sentence since it highlights inherent difficulties in determining environmental parameter impact on diazotrophs. (Pg. 32, lines 16-19).*

**P33L9: "Consistent…abundant"— rephrase, this statement is meaningless out of context.**
*We agree and have rephrased and merged it with the following sentence. (Pg. 33, lines 23-25).*

**P33L17-21: "According…tests"— remove this sentence.**
*We did not remove the sentence since we feel that our results are important to highlight that there was a disconnect in the detection. A similar result was reported in a qPCR study by Thompson et al. 2014 (one of the first to describe the UCYN-A2 symbiosis in detail) that found symbiont/host ratios of 0.2-11 during 3 days of sampling. The same possible limitations applied, which they also state, and hence we felt it was still valid to include this in our summary of conclusions. A sentence on this study has been added to the discussion. (Pg. 28 lines 7-8)*

[revised manuscript text omitted]